# GPR161–GLI3 repressor signaling at cilia directs apical constriction and cell fate during cranial neural tube closure

Eric R. Brooks[1,*,‡], Sun-Hee Hwang[2,*], Kevin A. White[2] and Saikat Mukhopadhyay[2,‡]

## ABSTRACT

Failure to close the cranial neural tube, known as exencephaly/anencephaly, is a lethal congenital defect. However, the mechanisms driving patterning and reshaping of the broad cranial neural folds are poorly understood. Loss of the primary cilium-localized G protein-coupled receptor GPR161 causes ectopic, excessive hedgehog signaling in the mouse neural tube and fully penetrant exencephaly. GPR161 promotes GLI3 transcriptional repressor (GLI3R) formation while preventing GLI2 transcriptional activator formation. Here, we studied the mechanisms underlying cranial closure in mice using a *Gpr161* mutant allelic series, epistasis between *Gpr161* knockout and GLI effectors, and *in toto* imaging of cell behavior. A functional non-ciliary *Gpr161* knock-in implicated GPR161 ciliary localization directly in initiation and maintenance of cranial closure. Furthermore, *Gli3R* expression, but not *Gli2* loss, rescued exencephaly in *Gpr161* knockout mice. GLI3R specifically restricted forebrain ventral floor plate expansion and mediated apical constriction in the lateral midbrain neural folds prior to closure. These results reveal metamere-specific, cilia-dependent hedgehog repression thresholds in control of spatially restricted gene expression and dynamic cell behavior during cranial closure. Targeted interventions increasing hedgehog repression could ameliorate regional cranial defects.

KEY WORDS: Cilia, Exencephaly, GLI, Repressor, Neural tube, GPR161, Mouse

## INTRODUCTION

Formation of the central nervous system begins with neural tube closure, a complex process converting a sheet of neuroepithelial cells into the closed tube that forms the basis of the brain and spinal cord (Nikolopoulou et al., 2017). Defects in this process are among the most common and deleterious human structural birth defects, occurring in ∼1:2000 pregnancies (Wallingford et al., 2013; Greene and Copp, 2014). These defects are not uniform, instead occurring at specific positions along the head-to-tail axis, indicating that unique genetic and cellular mechanisms are required in these distinct domains. Anencephaly, or the complete loss of brain tissue, is a defect caused by exencephaly, i.e. closure failure specifically in the cranial domain. These cranial-specific defects are peri-natal lethal and account for approximately one-third of reported human neural tube defects (Zaganjor et al., 2016; Avagliano et al., 2019).

During neural tube formation, closure initiates at multiple sites along the rostrocaudal axis, and closure finalizes by a zippering action between these closure points (Copp et al., 2003; Copp, 2005). In mice, 'closure 1' initiates at hindbrain/cervical boundary at around the 4-7 somite stage [approximately embryonic day (E) 8.25] driving final hindbrain and spinal closure. In contrast, 'closure 2' and 'closure 3' begin in the forebrain/midbrain and rostral forebrain extremity, respectively, at later stages (∼10 somites, E8.5) completing anterior and midbrain-hindbrain neuropore closure by E9.5 (Yamaguchi and Miura, 2013). Failures in these closures result in exencephaly. Defects in closure arise from errors in reshaping the tissue, including a failure to appropriately elevate the neural folds and deflect them inwards towards the midline. While the mechanisms during spinal closure, including cell shape remodeling and convergent extension, have received significant attention (Nikolopoulou et al., 2017), those driving cranial closure remain opaque.

Apical constriction is a crucial and conserved cell remodeling program whereby spatially delimited populations of epithelial cells collectively shrink their apical surface area relative to their basal surface area to promote tissue bending or buckling (Heisenberg and Bellaiche, 2013; Martin and Goldstein, 2014; Vijayraghavan and Davidson, 2017; Goodwin and Nelson, 2021). During neural tube closure, apical constriction shows differential dynamic properties between the spinal and cranial regions (Christodoulou and Skourides, 2022; Ampartzidis et al., 2023; Baldwin et al., 2024). Additionally, while spinal neural tube closure requires apical constriction at critical points in the tissue spanning tens of cell diameters, we recently showed that cranial closure requires a specific pattern of apical constriction in the prospective midbrain, whereby every cell between the midline and lateral edges of the cranial folds, a span encompassing approximately 100 cell diameters, undergoes apical constriction (Brooks et al., 2020). This is likely a necessary adaptation to the specific challenges posed by the cranial neural folds, which, in addition to being significantly wider than spinal tissues, begin as highly convex and outwardly curved structures compared to the initially flat morphology in spinal regions.

Multiple mouse mutants characterized by high hedgehog (HH) signaling exhibit cranial neural tube defects (Harris and Juriloff, 2007, 2010; Murdoch and Copp, 2010; Juriloff and Harris, 2018; Somatilaka et al., 2020) and we recently showed that inappropriately high HH signaling levels in lateral cells lead to defects in actomyosin organization and apical constriction, resulting in failure of neural fold elevation and closure (Brooks et al., 2020). Because HH activity is also a key factor in patterning neural precursor fates along the dorsoventral axis of the neural tube (Dessaud et al., 2008; Briscoe, 2009), this single signal is a crucial regulator of both patterning and morphogenesis. Cell fate patterning downstream of HH signaling

[1]Department of Molecular Biomedical Sciences, College of Veterinary Medicine, North Carolina State University, Raleigh, NC 27695, USA. [2]Department of Cell Biology, University of Texas Southwestern Medical Center, Dallas, TX 75390, USA.
*These authors contributed equally to this work

‡Authors for correspondence (eric.brooks@ncsu.edu; saikat.mukhopadhyay@utsouthwestern.edu)

E.R.B., 0000-0003-3159-8626; S.M., 0000-0003-4790-3090

relies on well-established mechanisms altering the graded activity of the downstream GLI factors that activate or repress transcription in response to local HH levels (Kopinke et al., 2021). However, how HH activity levels control apical constriction to pattern the tissue remodeling programs driving closure, and any reliance on specific aspects of GLI activity, is unknown.

Neural tube patterning by HH relies on the primary cilium, a microtubule-based dynamic appendage templated from the mother centriole (Goetz and Anderson, 2010; Anvarian et al., 2019). We previously demonstrated that the G protein-coupled receptor GPR161 is localized at the primary cilium and acts as an HH pathway repressor. *Gpr161* knockout (ko) mice are lethal by E10.5, show high HH signaling throughout the neural tube, and have exencephaly (Mukhopadhyay et al., 2013). *GPR161* variants are associated with human neural tube defects (Kim et al., 2019). Removing cilia blocks neural tube defects in many high HH signaling mutants, including *Gpr161* ko (Huangfu et al., 2003; Mukhopadhyay et al., 2013; Somatilaka et al., 2020). However, cilia-specific effectors of cranial neural tube closure are not well understood because of a lack of mouse models that separate cilia structure from HH signaling during patterning and morphogenesis.

The transcriptional output of the HH pathway is determined by an intricate balance between Gli transcriptional activators and repressors (Kopinke et al., 2021). GPR161 promotes the formation of GLI3R (a truncated repressor form of GLI3) (Mukhopadhyay et al., 2013) and prevents GLI2 activation (Hwang et al., 2021). Downstream effects from GPR161 loss can arise from either lack of GLI3R or GLI2 activation in a tissue-specific manner (Hwang et al., 2021, 2023). Lack of *Gli2* or *Gli3R* expression both restrict ventral progenitor expansion in spinal neural tube (Hwang et al., 2021, 2023). However, whether GLI2 and GLI3 have divergent roles in cranial closure, and whether specific activation or repression activity is required is unknown. GPR161 also localizes to periciliary endosomes (Mukhopadhyay et al., 2013; Pal et al., 2016) and both ciliary and extraciliary pools coordinate in determining GLI3R thresholds (Hwang et al., 2021); however, the roles of these distinct pools in cranial neural tube development is unclear.

Here, we define the precise roles of GLI2 and GLI3 in tissue remodeling during cranial neural fold elevation, the overall HH signaling status of these tissues, and the resulting effects on cell fate patterns by using mouse models targeting specific GLI activity states in the background of *Gpr161* mutants. Further, using high-resolution imaging and computational approaches we probe how the patterned apical constriction program that drives closure responds to differing HH activity states. Strikingly, we show that exencephaly in *Gpr161* mutants can be rescued specifically by transgenic activation of constitutive GLI3 repression. Our results identify metamere-specific requirements for GLI2 and GLI3 activity and demonstrate that GLI3 repressive activity is a crucial downstream element interpreting HH activity states to drive cranial closure and patterning.

## RESULTS
### *Gpr161* localization to cilia modulates cranial closure
*Gpr161* knockout (*Gpr161* ko or *Gpr161^ko/ko^*) embryos are embryonic lethal at E10.5 (Mukhopadhyay et al., 2013), but the specific structural defects in these embryos are unclear. To better characterize these embryonic phenotypes, we used scanning electron microscopy (SEM; Fig. 1A). Control embryos at E10.25 had closed cranial neural tubes and well-developed optic cups, while *Gpr161* ko embryos had fully penetrant exencephaly and missing optic cups. We next used an allelic series to compare *Gpr161* ko with *Gpr161^mut1^*, in which a knock-in variant at the endogenous locus produces a

GPR161 protein that is competent in cAMP signaling, but fails to appropriately localize to or transit through cilia (Hwang et al., 2021). *Gpr161^ko/mut1^* embryos are lethal at E13.5 and *Gpr161^mut1/mut1^* are lethal by E14.5 (Hwang et al., 2021). *Gpr161^ko/mut1^* embryos had fully penetrant exencephaly at all stages and completely lacked optic cups, indicating that GPR161 function in cilia is required for cranial neural tube closure (Fig. 1B, Fig. S1A). Conversely, almost all *Gpr161^mut1/mut1^* we recovered at E11 or before appeared to have completed initial closure (Fig. 1C, Table 1, Fig. S1B), with wider cranial neural tissues (Fig. S1C). This suggests that GPR161 extraciliary pools can at least partially compensate for loss of ciliary GPR161 function during closure. However, *Gpr161^mut1/mut1^* embryos recovered at later stages exhibited highly penetrant exencephaly (Table 1, Fig. S1D), suggesting a failure to maintain cranial neural tube closure as the tissues rapidly grow. *Gpr161^mut1/mut1^* embryos also developed grossly normal optic cups at E11 (Fig. 1C) despite exhibiting microphthalmia at later stages (Hwang et al., 2021). We previously demonstrated that extraciliary pools of GPR161 in *Gpr161^mut1/mut1^* show higher GLI3R levels than *Gpr161* ko at E9.5 but not at later stages (Hwang et al., 2021). Taken together, our results suggest that both initiation and maintenance of cranial neural tube closure is contingent upon GLI3R thresholds generated by GPR161 in both ciliary and extraciliary compartments.

### *Gli3R* expression but not *Gli2* loss rescues HH activity expansion and exencephaly in *Gpr161* knockout mutants
GLI2 and GLI3R are the major two cilia-generated effectors in the HH pathway (Kopinke et al., 2021) and GPR161 activity promotes GLI3R formation while preventing GLI2 activator generation (Mukhopadhyay et al., 2013; Hwang et al., 2021) (Fig. 2A). Ventral cell fate expansion in the spinal cord of *Gpr161* ko is rescued completely by *Gli2* loss and partially by *Gli3R* expression (Hwang et al., 2021, 2023), but the role of these specific Gli populations in cranial tissues is less clear. To test directly whether cranial closure defects upon GPR161 loss are GLI2 dependent, we introduced a *Gli2* ko allele (Mo et al., 1997) into the *Gpr161* ko background, using recombination to generate linked heterozygotes given these genes share a chromosome. *Gpr161; Gli2* double ko embryos survived until E12.75, beyond the lethality at E10.5 exhibited by *Gpr161* ko alone (Mukhopadhyay et al., 2013; Hwang et al., 2021). However, *Gpr161; Gli2* double ko embryos exhibited exencephaly at full penetrance, equivalent to *Gpr161* ko alone (Fig. 2B,C). Further, whole-mount RNA *in situ* hybridization (ISH) revealed high levels of the HH pathway target *Ptch1* in the lateral folds of the open cranial neural tube in both *Gpr161* ko and *Gpr161; Gli2* double ko embryos (Fig. 2D). Thus, unlike the spinal cord, lack of *Gli2* neither restricts excessive HH signaling upon *Gpr161* loss, nor rescues cranial closure.

We next tested whether restoring GLI3R activity prevented exencephaly in the *Gpr161* ko background. To do so, we chose a recently described potent conditional knock-in allele of *Gli3* repressor (*Gli3^Δ701C^*) (Cao et al., 2013), which unlike the other commonly used *Gli3* repressor allele (*Gli3^Δ699^*) (Bose et al., 2002), recapitulates the *Shh* ko limb phenotypes (Litingtung et al., 2002; te Welscher et al., 2002) and exhibits heterozygotic embryonic lethality when ubiquitously expressed (*Gli3^Δ701^*). To mitigate lethality, mice with conditional *Gli3^Δ701C/Δ701C^* alleles were crossed with *CAG-Cre* (Sakai and Miyazaki, 1997) along with *Gpr161^ko^* or *Gpr161^flox^* alleles to generate *Gli3R^Δ701^*-expressing embryos in a *Gpr161* ko background. *Gpr161* ko; *Gli3R^Δ701/+^* embryos survived until E12.0, and, strikingly, the majority of these embryos exhibited both fully closed cranial tubes and well-developed optic cups (Fig. 3) and this rescue effect held across multiple litters and stages of mid-embryonic

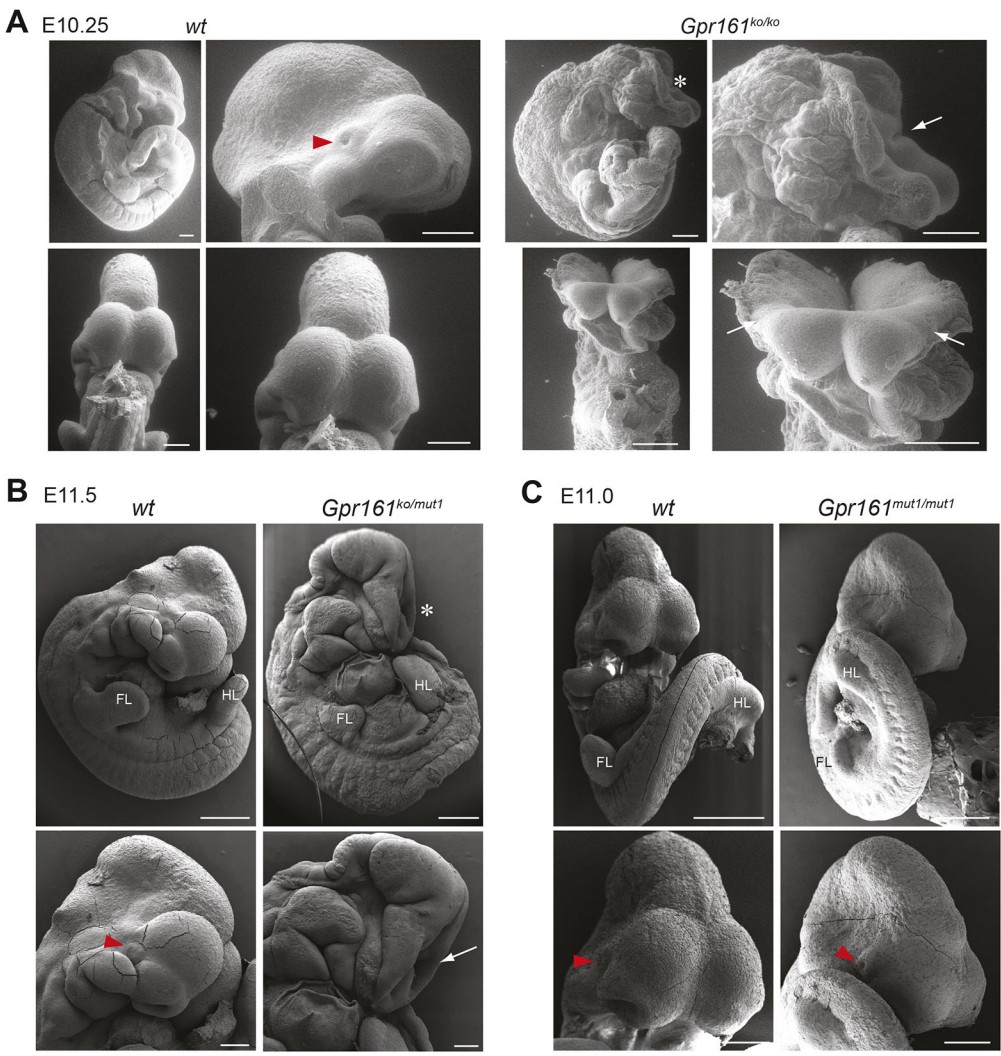

**Fig. 1. Exencephaly in *Gpr161* ko mice is cilia regulated.** (A) SEM images of control wild-type (*wt*) and *Gpr161^{ko/ko}* embryos at E10.25. Top left and middle panels, lateral views. Top right and bottom panels, *en face* views. Note exencephaly and lack of optic cups in *Gpr161^{ko/ko}*. (B) SEM images of representative control wild-type (*wt*) (*n*=2) and *Gpr161 ko/mut1* (*n*=3) littermate embryos at E11.5. Note exencephaly and small forelimbs in *Gpr161^{ko/mut1}* embryos. (C) SEM images of representative control (*n*=3) and *Gpr161^{mut1/mut1}* (*n*=3) littermate embryos at E11. Note mostly closed cranial neural tube in *Gpr161^{mut1/mut1}* embryos at this stage despite structurally widened anterior neural tube (see Fig. S1). Examples of exencephaly in *Gpr161^{mut1/mut1}* embryos are shown in Fig. S1. Exencephaly is marked by asterisks. Open cranial folds are marked by arrows. Optic cups are marked by red arrowheads. FL, forelimb bud; HL, hindlimb bud. Scale bars: 500 µm (A, whole embryo images; B,C, top); 200 µm (A, zoomed panels; B,C, bottom). See also Fig. S1.

development (Table 2). In *Gpr161* ko; *Gli3R^{Δ701/+}* embryos showing only partial rescue, exencephaly persisted at the level of the midbrain-hindbrain neuropore, but the optic cups were well developed and the rostral most neuropore had closed (Fig. 3B, Table 2). The incomplete rescue might stem from early variability in expression due to the *CAG-Cre* driver, with lower *Gli3R* levels not meeting the threshold required to fully rescue midbrain-hindbrain closure. In comparison, *Gpr161^{ko/mut1}* mutants, which had fully penetrant exencephaly and fully open cranial neural tubes, did not have optic cups (Fig. 1B, Fig. S1A). Together, our data indicate that *Gli3R* expression, but not *Gli2* loss, can rescue exencephaly induced by the excessive activation of HH signaling in the *Gpr161* ko background. These results indicate that de-repression, and not excessive activation, of HH pathway targets underlies cranial closure defects.

We next tested the HH pathway signaling status in *Gpr161* ko; *Gli3R^{Δ701/+}* embryos compared to control and *Gpr161* ko alone embryos at E9.25 using whole-mount *Ptch1* ISH followed by

horizontal cryosectioning. Interestingly, the abnormally high levels of *Ptch1* observed in the forebrain and hindbrain regions of *Gpr161* ko were reverted in *Gpr161* ko; *Gli3R^{Δ701/+}* embryos (Fig. 4A). At later stages (~E10.75), ISH for *Ptch1* showed normal expression in the closed cranial neural tube region of the fully rescued *Gpr161* ko; *Gli3R^{Δ701/+}* embryos, although the frontonasal prominences still retained elevated *Ptch1* levels (Fig. 4B), possibly from increased GLI2 activator activity (Chang et al., 2016). The partially rescued *Gpr161* ko; *Gli3R^{Δ701/+}* embryos showed restored optic cups and closed rostral-most cranial neural tube at the anterior neuropore despite high levels of *Ptch1* in the open portion of the cranial folds as well as widely separated frontonasal prominences (Fig. 4B). Thus, *Gli3R* expression rescued exencephaly and restricted high HH signaling in the open cranial neural tube of *Gpr161* ko. Together, these data indicate that both elevated HH target response and cranial closure defects stem from GLI3 target de-repression rather than an overactivation of GLI2 target response.

**Table 1. Quantification of exencephaly phenotypes in *Gpr161^{mut1/mut1}* embryos at different stages of mid-embryonic development**

| | Number of litters | Total number of embryos | Number of *mut1/mut1* embryos | Number of *mut1/mut1* embryos with exencephaly | Exencephaly occurrence in *mut1/mut1* embryos (%) |
|---|---|---|---|---|---|
| E9.5-E9.75 | 9 | 70 | 11 | 1 | 9.1 |
| E10.25-E11.0 | 8 | 65 | 20 | 3 | 15.0 |
| E12.5-E13.5 | 10 | 66 | 14 | 9 | 64.3 |

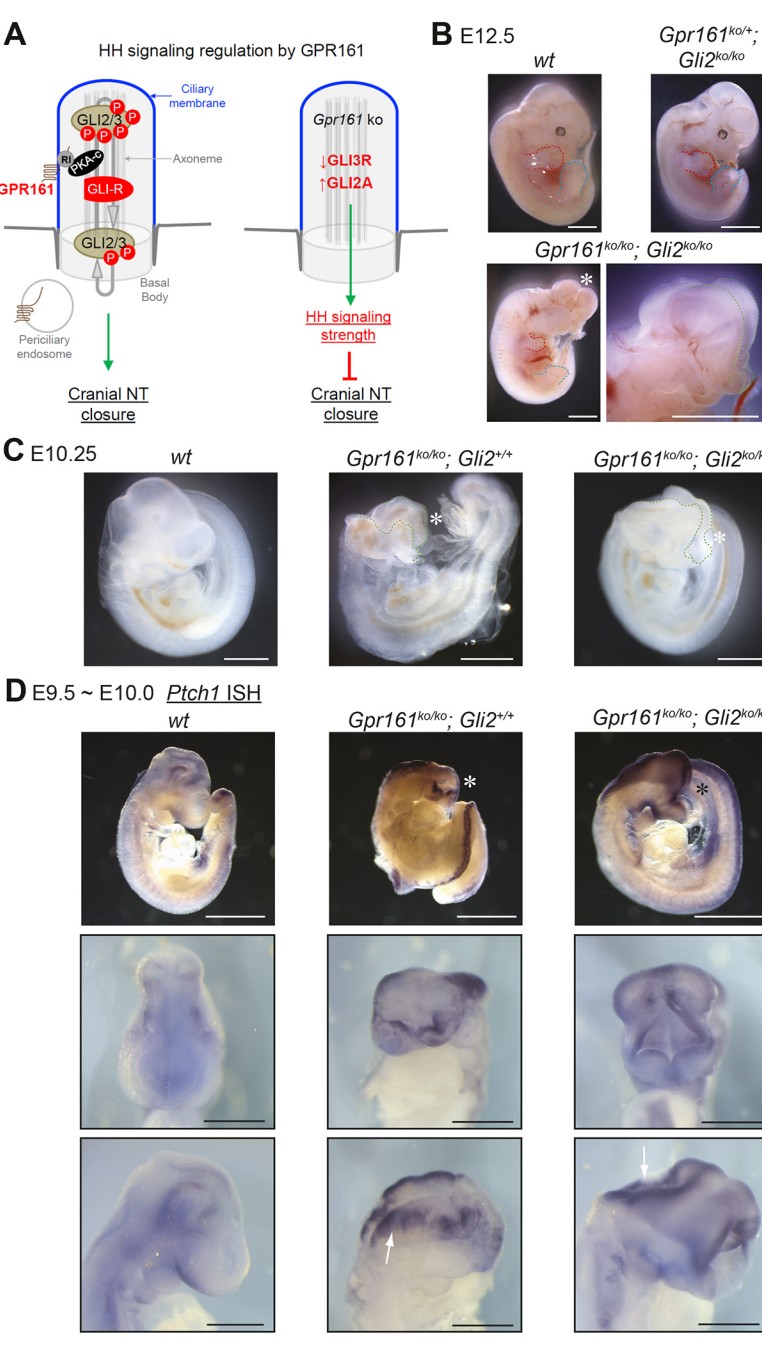

**Fig. 2. Exencephaly and high HH signaling in cranial neural tube in *Gpr161* ko is not rescued by *Gli2* loss.** (A) Schematic model of cilia-dependent HH signal modulation in wild-type (left) and *Gpr161ko/ko* (right) conditions. GPR161 directly binds to the PKA RIα subunit (RI) in cilia and also localizes to periciliary endosomes. Loss of *Gpr161* leads to both a strong decrease in the formation of the repressor form of GLI3 (GLI3R) and a modest increase in the formation of activator GLI2 (GLI2A). Together, these changes to GLI factor populations lead to both an overactivation and de-repression of downstream HH target genes. (B) Whole-mount images of control wild-type (*wt*), *Gli2ko/ko* and *Gpr161ko/ko; Gli2ko/ko* littermates at E12.5. Note exencephaly in *Gpr161ko/ko; Gli2ko/ko* littermates. *Gli2ko/ko* (*n*=6) and *Gpr161ko/ko; Gli2ko/ko* (*n*=5) from multiple litters. Enlarged view of the head region is shown for *Gpr161ko/ko; Gli2ko/ko*. Open neural folds are marked by green dashed lines. Red and blue dashed lines mark forelimb and hindlimb buds, respectively. (C) Whole-mount images of control, *Gpr161ko/ko* and *Gpr161ko/ko; Gli2ko/ko* at E10.25. Control and *Gpr161ko/ko; Gli2ko/ko* are littermates. *Gpr161ko/ko* (*n*>23) and *Gpr161ko/ko; Gli2ko/ko* (*n*=6) from multiple timed pregnancies. Note exencephaly in *Gpr161ko/ko* and *Gpr161ko/ko; Gli2ko/ko* littermates. Open neural folds are marked by green dashed lines. (D) Whole-mount *Ptch1 in situ* hybridization in control, *Gpr161ko/ko* and *Gpr161ko/ko; Gli2ko/ko* at E9.5-E10. Top, lateral views; middle, enlarged *en face* views; bottom, enlarged lateral views. Note exencephaly in *Gpr161ko/ko* and *Gpr161ko/ko; Gli2ko/ko* and high *Ptch1* levels in lateral neural folds marked by white arrows. Exencephaly is marked by asterisks. Scale bars: 2 mm (B); 1 mm (C; D, top); 500 µm (D, middle and bottom).

## *Gli3R* expression but not *Gli2* loss restores FOXA2-dependent floor plate progenitors in *Gpr161* knockout forebrain

Our data demonstrate that *Gli3R* expression, but not *Gli2* loss, specifically restricted high HH signaling in the open cranial neural tube of *Gpr161* ko, contrary to what we observed in studies of the spinal cord (Hwang et al., 2023). To determine whether these region-specific differences reflected broader differences in HH response along the anteroposterior axis or instead differences at the level of specific metameres, we compared neural patterning in the *Gpr161* ko background after deletion of *Gli2* or global activation of *Gli3* repression, in horizontal sections of E9.25 embryos. Sections from control embryos at the level of the prospective optic cups showed full forebrain closure, but the forebrain tissues were open and the optic cups missing in equivalent *Gpr161* ko sections. Both

closure and optic cup morphology were rescued in *Gpr161* ko; *Gli3R* embryos, whereas *Gpr161* ko; *Gli2* showed no rescue in either phenotype (Fig. 5A, Figs S2, S3). The hindbrain at this stage started to close in all *Gpr161* ko and *Gpr161; Gli2* double ko embryos.

We observed extensive ventralization of *Gpr161* ko forebrain tissues when compared to controls. Floor plate progenitors expressing FOXA2, pMN progenitors expressing OLIG2, and p3/pMN/p2 progenitors expressing NKX6-1 all showed enlarged expression domains over controls (Fig. 5A,B, Fig. S2). The expansion of these markers, including the ventral-most floor plate marker FOXA2, in *Gpr161* ko; *Gli3RΔ701/+* was greatly rescued, becoming equivalent to those of *Gli3RΔ701/+* only-expressing embryos. Loss of *Gli2* alone reduced FOXA2 levels compared to controls as reported for the spinal cord (Bai and Joyner, 2001), but the FOXA2 expansion induced by *Gpr161* loss was not restored

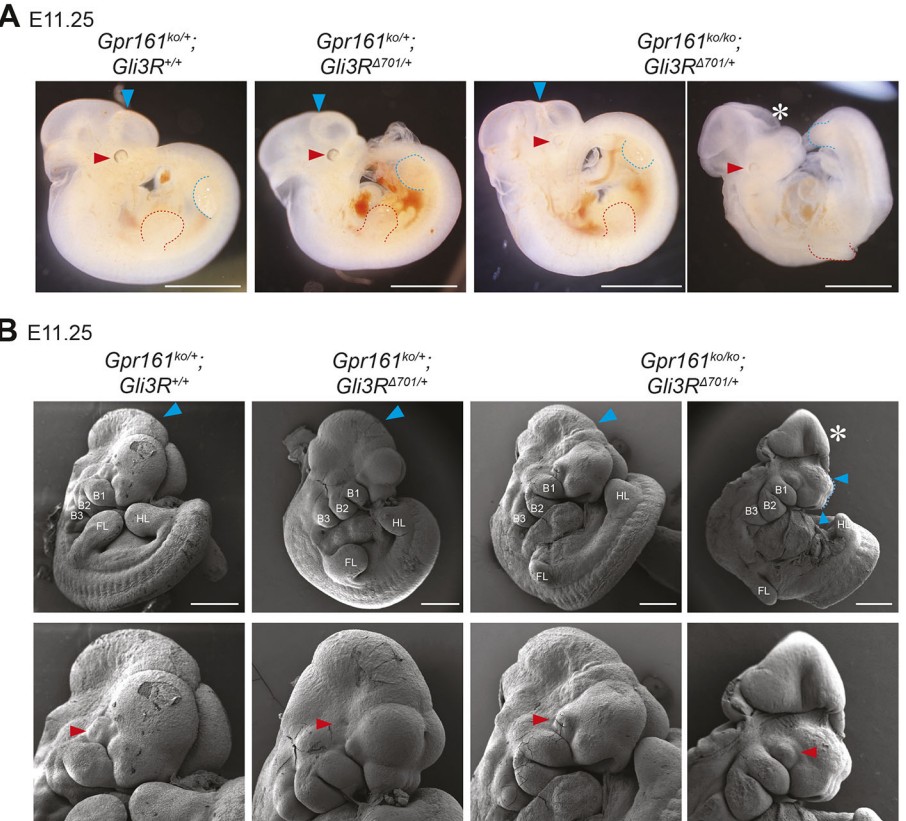

**Fig. 3. Exencephaly in *Gpr161* ko is rescued by *Gli3R* expression.** (A) Whole-mount images of control, *Gli3R^{Δ701}* and *Gpr161^{ko/ko}; Gli3R^{Δ701/+}* littermates at E11.25. Red and blue dashed lines mark forelimb and hindlimb buds, respectively. (B) SEM images of control, *Gli3R^{Δ701}* and *Gpr161^{ko/ko}; Gli3R^{Δ701/+}* littermates at E11.25 (41-42 somites). Closed cranial neural tubes and optic cups in all genotypes are marked by blue and red arrowheads, respectively. Exencephaly is marked by asterisks. Scale bars: 2 mm (A); 500 µm (B, top); 200 µm (B, bottom). B1, B2, B3, branchial arches 1-3; FL, forelimb bud; HL, hindlimb bud.

upon simultaneous deletion of *Gli2* in forebrain tissues, including in the anterior-most closed regions (Fig. 5A,B, Fig. S2). OLIG2 expansion was also not rescued in *Gpr161; Gli2* double ko forebrain regions compared to *Gpr161* ko. Intriguingly, the lack of the dorsolateral marker PAX6 in *Gpr161* ko forebrain was restored in the dorsoventral extent by either *Gli3R^{Δ701/+}* or *Gli2* deletion, although the optic cups, which are also marked by PAX6 expression, were present in *Gpr161* ko; *Gli3R^{Δ701/+}* but absent in *Gpr161; Gli2* double ko.

In striking contrast to the forebrain, the hindbrain regions of the same embryos showed restored ventral restriction of floor plate marker FOXA2 to wild-type levels in *Gpr161* ko; *Gli3R^{Δ701/+}* and showed further restriction in *Gpr161; Gli2* double ko embryos (Fig. 5A,C, Fig. S3), similar to our prior observations in the spinal cord (Hwang et al., 2023). Similarly, ventral restriction of NKX6-1 and lack of expression of the dorsolateral marker PAX6 in *Gpr161* ko was partially restored in both *Gpr161* ko; *Gli3R^{Δ701/+}* and

*Gpr161; Gli2* double ko hindbrains (Fig. 5A,C, Fig. S3), as we previously reported for the spinal cord (Hwang et al., 2023).

These data reveal differences in pathway logic between the forebrain and hindbrain/spinal cord that appears to depend exclusively on GLI3 repression to prevent ventralization, and hindbrain and spinal cord, where either GLI3R expression or GLI2 absences can rescue ventralization (Hwang et al., 2023) (Fig. 5D). These results are consistent with our previous finding that excessive HH signaling induces metamere-specific gene expression changes (Brooks et al., 2025), and support a model for region-specific downstream requirements in fully articulating the patterned response to HH activity levels.

### *Gli3R* expression restricts high HH signaling at cranial neural fold elevation stages of *Gpr161* ko

We next examined the HH pathway signaling status in *Gpr161* ko and *Gpr161* ko; *Gli3R^{Δ701/+}* embryos during earlier cranial neural

**Table 2. Quantification of phenotypic rescue in *Gpr161^{ko/ko}; Gli3R^{Δ701/+}* embryos at different stages of mid-embryonic development**

| | Litters | Total number of embryos | Number of *Grp161^{ko/ko}; Gli3R^{Δ701/+}* embryos | Number of *Grp161^{ko/ko}; Gli3R^{Δ701/+}* embryos rescued (% live embryos)* | Number of *Grp161^{ko/ko}; Gli3R^{Δ701/+}* embryos partially rescued | Number of *Grp161^{ko/ko}; Gli3R^{Δ701/+}* embryos dead |
|---|---|---|---|---|---|---|
| E9.25-E9.5 | 9 | 60 | 12 | 12 (100%) | 0 | 0 |
| E9.75-E10.25 | 4 | 28 | 6 | 6 (100%) | 0 | 0 |
| E10.5-E11.0 | 5 | 38 | 10 | 5 (55.6%) | 4 | 1 |
| E11.25-E11.5 | 5 | 32 | 8 | 6 (85.7%) | 1 | 1 |
| E12.0-E13.5 | 13 | 85 | 6 | 0 | 0 | 6 |

*Phenotypic rescue defined as closed cranial tube; lack of rescue defined as persistent exencephaly, although optic cups were formed. Similar rescue of exencephaly was observed in *Gpr161^{ko/ko}; Gli3R^{Δ701/+}* male and female embryos, e.g. among the E10.5-E11.5 litters, 5/11 rescued embryos were males, and 3/5 partially rescued were males.

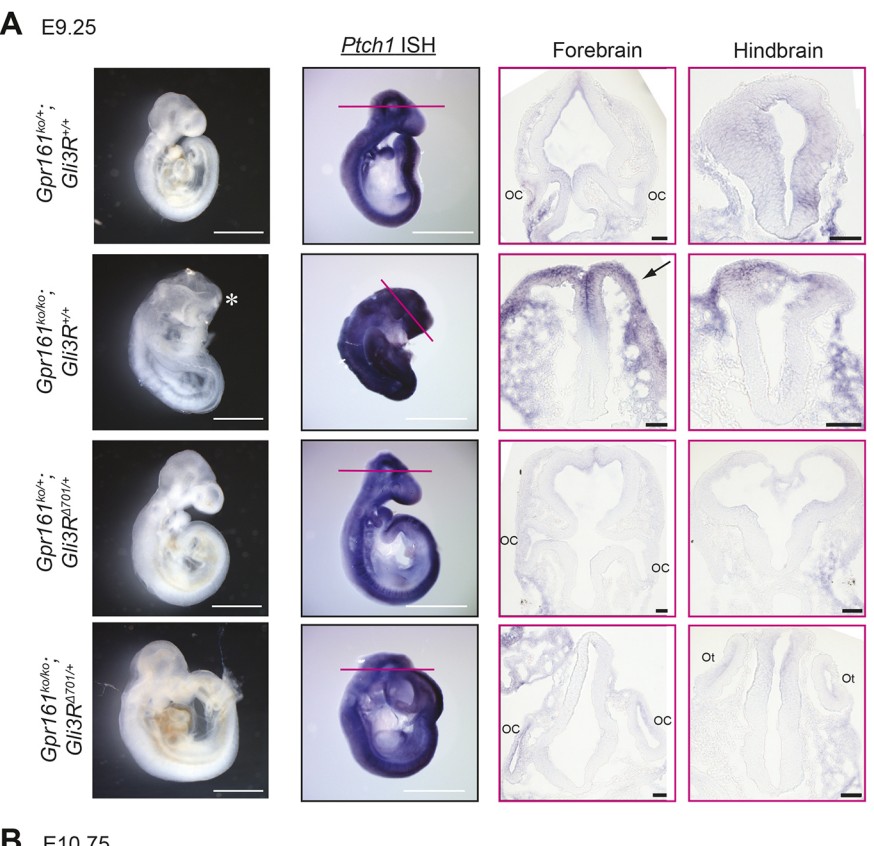

**Fig. 4. High HH signaling and open cranial neural tube in *Gpr161* ko embryos is rescued by *Gli3R* expression.** (A) Whole-mount images of *Ptch1 in situ* hybridization in littermates at E9.25 (20-22 somites). Genotypes of littermates: control (*n*=2), *Gli3R*$^{Δ701/+}$ (*n*=1), *Gpr161*$^{ko/ko}$ (*n*=1) and *Gpr161*$^{ko/ko}$; *Gli3R*$^{Δ701/+}$ (*n*=2). Horizontal sections for forebrain and hindbrain at the designated levels (pink lines) shown to the right. Note open forebrain cranial folds (arrow) and closed hindbrain in *Gpr161*$^{ko/ko}$ showing high *Ptch1* levels dorsolaterally that was rescued in *Gpr161*$^{ko/ko}$; *Gli3R*$^{Δ701/+}$ littermates. OC, optic cups; Ot, otic capsule. (B) Whole-mount images of *Ptch1 in situ* hybridization in control and *Gpr161*$^{ko/ko}$; *Gli3R*$^{Δ701/+}$ at E10.75. Note exencephaly from open midbrain-hindbrain neuropore in partially rescued *Gpr161*$^{ko/ko}$; *Gli3R*$^{Δ701/+}$ with the closed anterior neuropore marked by arrowheads. Blue arrowheads indicate closed anterior neural tube. Asterisk marks exencephaly. Scale bars: 1 mm (A, whole embryo images); 50 µm (A, sections); 1 mm (B).

fold elevation stages by performing *Ptch1* ISH at E8.25. *Ptch1* levels provide a relatively robust marker for HH signaling status at these stages, compared to ventral progenitor markers, which are more variable (Mukhopadhyay et al., 2013). Embryos at this stage are unturned, as axial rotation from lordotic to flexed fetal position initiates around ~E8.5 (9 somites) (Poelmann et al., 1987). Compared to controls, the cranial folds of *Gpr161* ko embryos at these stages showed higher *Ptch1* expression (Fig. 6A), whereas *Gli3R*$^{Δ701/+}$ alone embryos showed reduced ventral *Ptch1*. At mid-elevation stages, *Gpr161* ko; *Gli3R*$^{Δ701/+}$ embryos showed a rescue of *Ptch1* levels (Fig. 6A). However, in younger embryos (<6 somites) *Gpr161* ko did not show significantly elevated *Ptch1* expression (Fig. 6B) making potential rescue difficult to judge. Thus, *Gli3R* expression restricted high HH signaling induced by *Gpr161* by mid-elevation stages at the latest.

### *Gpr161* ko mutants show failure in lateral cell apical constriction in the future midbrain, but not forebrain

We next examined whether *Gpr161* ko exhibited defective cell remodeling during cranial neural fold elevation using *in toto* confocal microscopy and computational image segmentation of embryos at mid-elevation stages (5-7 somites; Materials and Methods). As predicted, cells from the putative midbrain of *Gpr161* ko embryos showed significantly enlarged apical cell areas compared to controls

(Fig. 7A-D,G). These defects occurred in the absence of changes in lateral cell height (Fig. 7N-P), and we previously showed that modulation of HH has no impact on cell proliferation rates at these stages (Brooks et al., 2020). These data reveal a failure of apical constriction within the lateral constrictive domains of *Gpr161* ko mutants, indicating that altered HH activity, not changes in ciliary architecture, is the driver of these defects. We next investigated midbrain midline cells, which are exposed to the highest activity of HH signaling and are uniquely short and apically expanded in wild-type tissues. While these cells exhibit wild-type morphology in mutants with strong ectopic or expanded HH signaling, they become inappropriately tall with reduced surface areas in mutants with reduced midline HH activity (Brooks et al., 2020). Consistent with strong global HH activation, midline cells in *Gpr161* ko mutants were indistinguishable from controls in both apical area (Fig. 7E,F,H) and apicobasal height (Fig. 7N-P). Because *Gpr161* mutants showed differential patterning consequences later along the cranial neuraxis, we next investigated whether lateral apical constriction defects were also regionalized by examining cell morphology in the putative forebrain. Surprisingly, cells from the lateral forebrain in both control and *Gpr161* ko embryos showed average apical areas and distributions similar to those of control midbrain tissues, indicating that there is no forebrain apical remodeling defect in mutants at these stages (Fig. 7I-M). Thus, the impact of HH signaling on apical

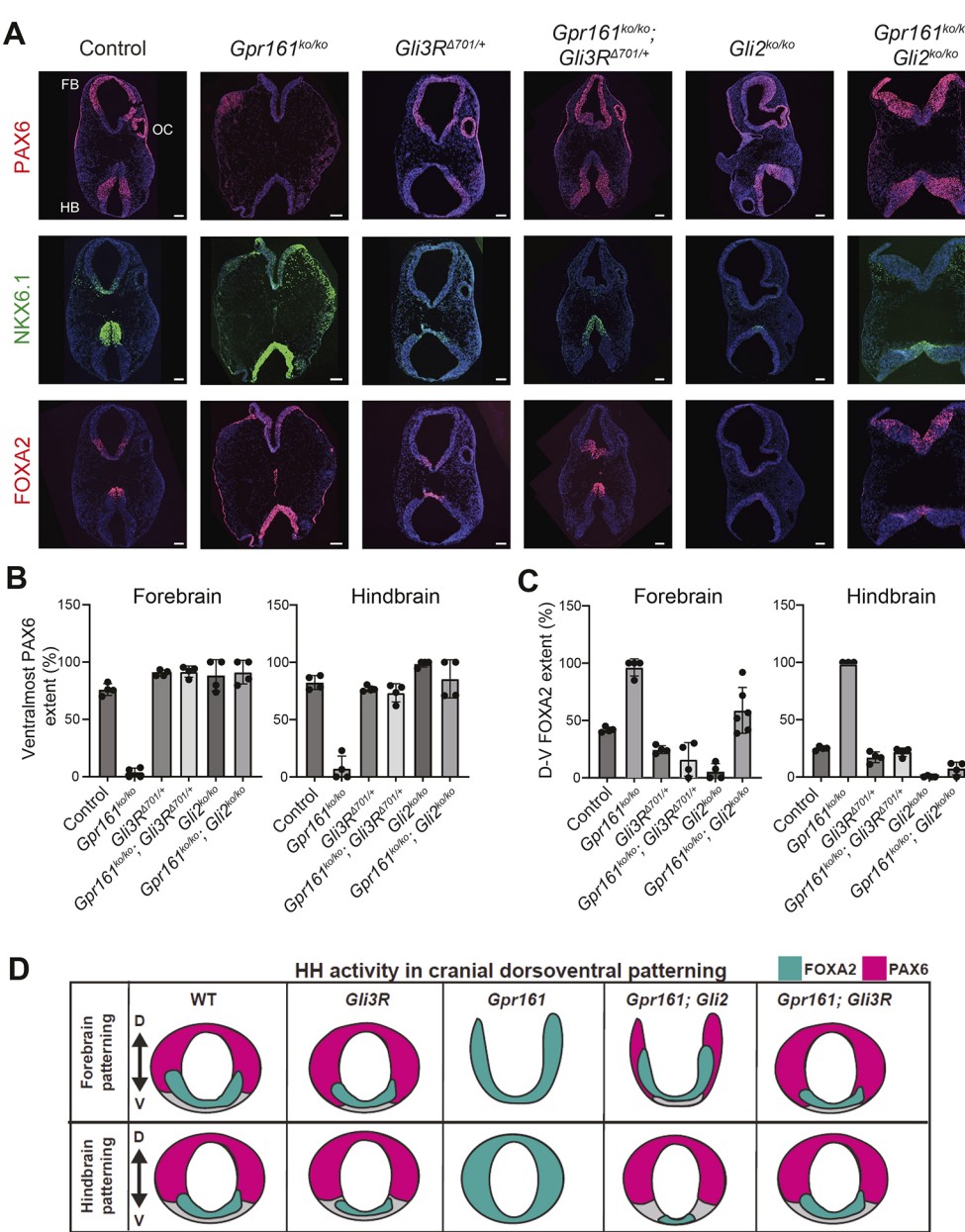

**Fig. 5. Floor plate marker ventralization in E9.25 *Gpr161* ko forebrain is selectively rescued by *Gli3R* expression but not from *Gli2* deletion.** (A) Forebrain (top) and hindbrain (bottom) cranial horizontal sections immunostained using designated markers from embryos dissected at E9.25 (18-22 somites, except Gli2 ko/ko at 25 somites) of the following genotypes: wild-type (*n*=6), *Gpr161*^{ko/ko} (*n*=3), *Gli3R*^{Δ701/+} (*n*=6), *Gpr161*^{ko/ko}; *Gli3R*^{Δ701/+} (*n*=3), *Gli2*^{ko/ko} (*n*=6), *Gpr161*^{ko/ko}; *Gli2*^{ko/ko} (*n*=6). The same sections for each genotype were co-stained for FOXA2 and NKX6-1 and consecutive sections for PAX6. All images are counterstained with DAPI. Scale bars: 100 μm. FB, forebrain; HB, hindbrain; OC, optic cup. See also Figs S2 and S3, which show magnified forebrain and hindbrain images of the same sections (except for staining of separate closed hindbrain sections for *Gpr161*^{ko/ko}; *Gli3R*^{Δ701/+} and *Gpr161*^{ko/ko}; *Gli2*^{ko/ko} and additional staining for OLIG2 for all genotypes). Additional anterior to posterior sections in *Gpr161*^{ko/ko}; *Gli2*^{ko/ko} brain are shown in Fig. S2. (B,C) Quantification of FOXA2 dorsoventral (D-V) extent and ventral-most PAX6 extent with respect to full extent of the neural tube in forebrain and hindbrain regions. All data shown as mean±s.d. Note forebrain FOXA2 ventralization in *Gpr161*^{ko/ko} rescued by *Gli3R* expression only, whereas hindbrain FOXA2 ventralization in *Gpr161*^{ko/ko} is rescued by *Gli3R* expression or *Gli2* deletion. Ventral-most PAX6 extents are not impacted from ventralized FOXA2 levels in *Gpr161*^{ko/ko}; *Gli2*^{ko/ko}. (D) Summary of the regionalized impacts on dorsoventral neural precursor patterning. WT, wild type.

constriction blockade appears to be limited to midbrain tissues, but this region-specific defect is sufficient to block closure along the majority of the cranial neuraxis.

### Expression of the *Gli3R* transgene is sufficient to rescue midbrain lateral apical constriction and exencephaly

We next wanted to understand the pathway logic governing the role of HH in apical constriction and cranial closure in more detail. In addition to the failure of exencephaly rescue in *Gpr161* ko; *Gli2* ko double mutants, we previously demonstrated that removing *Gli2* alone did not block lateral apical constriction in the midbrain (Brooks et al., 2020), indicating that positive HH target regulation is unlikely to underlie these defects. We therefore investigated whether lateral apical constriction defects resulted from a failure to establish effective GLI3 repression in the lateral domain. To do so, we compared control and *Gpr161* ko; *Gli3R*^{Δ701} double mutants at the 5-7 somite stages.

*Gli3R*^{Δ701/+} expression alone had no significant impact on the average apical area of lateral midbrain cells compared to controls

(Fig. 8), although it did create subtle changes in the distribution of cell areas, with a slight increase in highly constricted cells at the expense of cells with moderate cell area (Fig. 8, Fig. S4). As discussed above, *Gpr161* ko mutants showed a significant increase in the average apical area of lateral cells, compared to both control and *Gli3R* embryos (Fig. 8). This increase in area was driven by a strong reduction in the proportion of the most constricted cells and a corresponding increase in cells with very large apical areas (Fig. 8, Fig. S4). Conversely, *Gpr161* ko; *Gli3R*^{Δ701/+} embryos showed a strong rescue of lateral apical constriction, with no significant differences observed in the average apical area between these compound mutants, and either control or *Gli3R*^{Δ701/+} embryos. Rescue appeared to be driven primarily by a reversion of the number of highly constricted cells back to control levels and a decrease in cells displaying abnormally large apical areas (Fig. 8, Fig. S4). Thus, the repressive activity of GLI3 is specifically required to appropriately pattern apical constriction within the tissue and promote cranial neural tube closure.

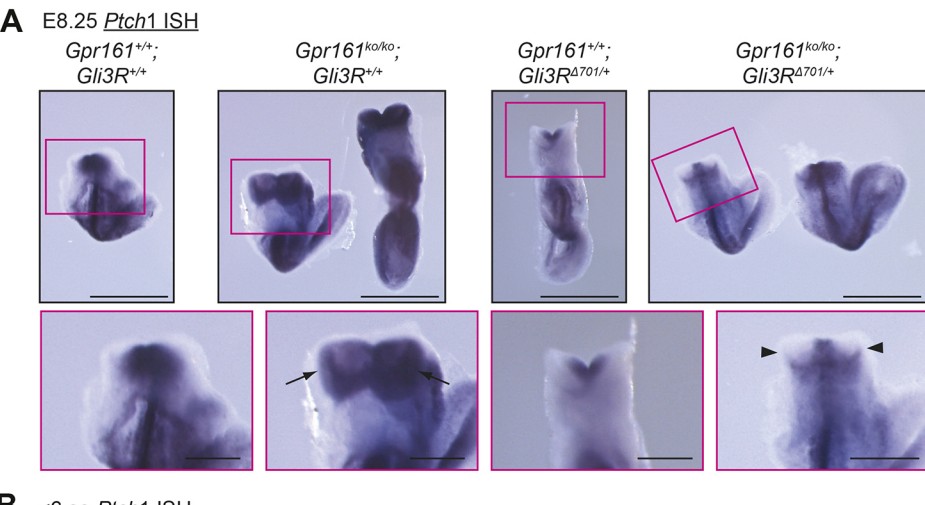

**Fig. 6. High HH signaling in *Gpr161* ko is rescued by *Gli3R* expression during cranial fold elevation.** (A) Whole-mount images of *Ptch1 in situ* hybridization in control (*n*=3), *Gpr161*$^{ko/ko}$ (*n*=3), *Gli3R*$^{\Delta701/+}$ (*n*=1) and *Gpr161*$^{ko/ko}$; *Gli3R*$^{\Delta701/+}$ (*n*=2) at E8.25 prior to axial rotation (<8 somites). Boxed regions are shown at higher magnification below. Arrows mark *Ptch1* in lateral cranial folds in *Gpr161*$^{ko/ko}$ and arrowheads mark absent *Ptch1* in lateral cranial folds in *Gpr161*$^{ko/ko}$; *Gli3R*$^{\Delta701/+}$ embryos. (B) Whole-mount images of *Ptch1 in situ* hybridization in smaller control (*n*=2) and *Gpr161*$^{ko/ko}$ littermates at E8.25 [<6 somites (so)] (*n*=5). Boxed regions are shown at higher magnification on the right. Note lack of *Ptch1* increase in lateral cranial folds in *Gpr161*$^{ko/ko}$ at this stage. Scale bars: 500 μm (A, top); 200 μm (A, bottom; B).

## DISCUSSION

In this study, we explore the role of HH signaling and its downstream transcriptional consequences in coordination of two major overlapping aspects brain development: early cranial neural tube closure and neural precursor patterning. By altering GLI activity in specific ways in the background of excessive and ectopic HH signaling driven by *Gpr161* loss, we show that activation and de-repression of the HH pathway have separable outcomes in both processes. Removal of GLI2, which acts as the predominant pathway activator and is required to establish the ventral cell fates arising in cells with the highest HH response in the spinal cord (Mo et al., 1997; Matise et al., 1998; Bai and Joyner, 2001), was unable to effectuate either the rescue of expanded ventral neural precursor fates in the forebrain or cranial closure defects. Conversely, transgenic expression of a constitutively repressive variant of GLI3 showed a stronger rescue of overall dorsal-ventral fate specification in cranial tissues, as well as a strong rescue of cranial neural tube closure. Furthermore, constitutive *Gli3R* expression, but not *Gli2* loss (Brooks et al., 2020), mediated apical constriction in the lateral midbrain neural folds prior to closure. While GPR161 has been previously linked with the WNT pathway (Li et al., 2015; Kim et al., 2023) and the planar cell polarity effector FUZ (Kim et al., 2024) in spina bifida, our experiments here demonstrate that GPR161-mediated HH pathway regulation controls cranial neural tube closure. Together, our data indicate that specific loss of repression of HH targets is a major factor in patterning cranial neural precursor fate and cell shape remodeling during cranial closure.

### Cilia-generated GLI3R thresholds in cranial neural tube closure and maintenance

To our knowledge, this is the first time that an exencephaly phenotype has been rescued by a cilium-generated effector (GLI3R) while maintaining intact cilia. We find two instances, in *Gpr161*$^{mut1/mut1}$ and in some *Gpr161* ko; *Gli3R* embryos, where initial closure occurs but maintenance at the midbrain-hindbrain neuropore is affected (Fig. 9A,B). In both genotypes, we note closure of the anterior neuropore despite an open midbrain-hindbrain neuropore at later embryonic stages. These results suggest a potential role of GLI3R levels in not only ensuring proper initial closure, but also in maintaining this closed state as the cranial tissues continue to expand and undergo further morphogenesis. The defective maintenance of closed cranial neural tissues could come from a failure to appropriately specify the lateral borders of the neural plate, which drive neural fold fusion (Pyrgaki et al., 2011; Kimura-Yoshida et al., 2015; Ray and Niswander, 2016; Nikolopoulou et al., 2019). Alternatively, the significant differences in the geometry of the cranial tissues after closure found in *Gpr161*$^{mut1/mut1}$ embryos (Hwang et al., 2021) could interfere with the biomechanical process of zippering (Maniou et al., 2021) and affect tissue deformability post-closure (McLaren et al., 2025).

It is also notable that some *Gpr161; Gli3R* embryos showed a partial rescue of forebrain closure, but the midbrain/hindbrain remained open, raising the possibility that higher GLI3R thresholds might be required to ensure this more posterior closure. Given that we also previously reported that GPR161 activity regulates GLI3R threshold-specific morphophenotypic outcomes in other tissues (Hwang et al., 2023), cross-talk between ciliary and extraciliary pools, mediated by GPR161, which localizes to both ciliary and periciliary endosomes (Mukhopadhyay et al., 2013; Pal et al., 2016), may be crucial for modulation of tissue-specific GLI3R levels. Such a separation may lead to additional regulatory power, as emerging evidence suggests that specialized mechanisms drive GPR161 activity in the two populations: ciliary GPR161 activity is promoted by A-kinase anchoring protein (AKAP) activity (Bachmann et al., 2016) (Fig. 2A), whereas extraciliary GPR161 could undergo allosterically regulated Gαs-coupled cAMP generation (Hoppe et al., 2024). Further supporting this split activity model, our data suggest that excess extraciliary pools can at least partially compensate for lack of ciliary function

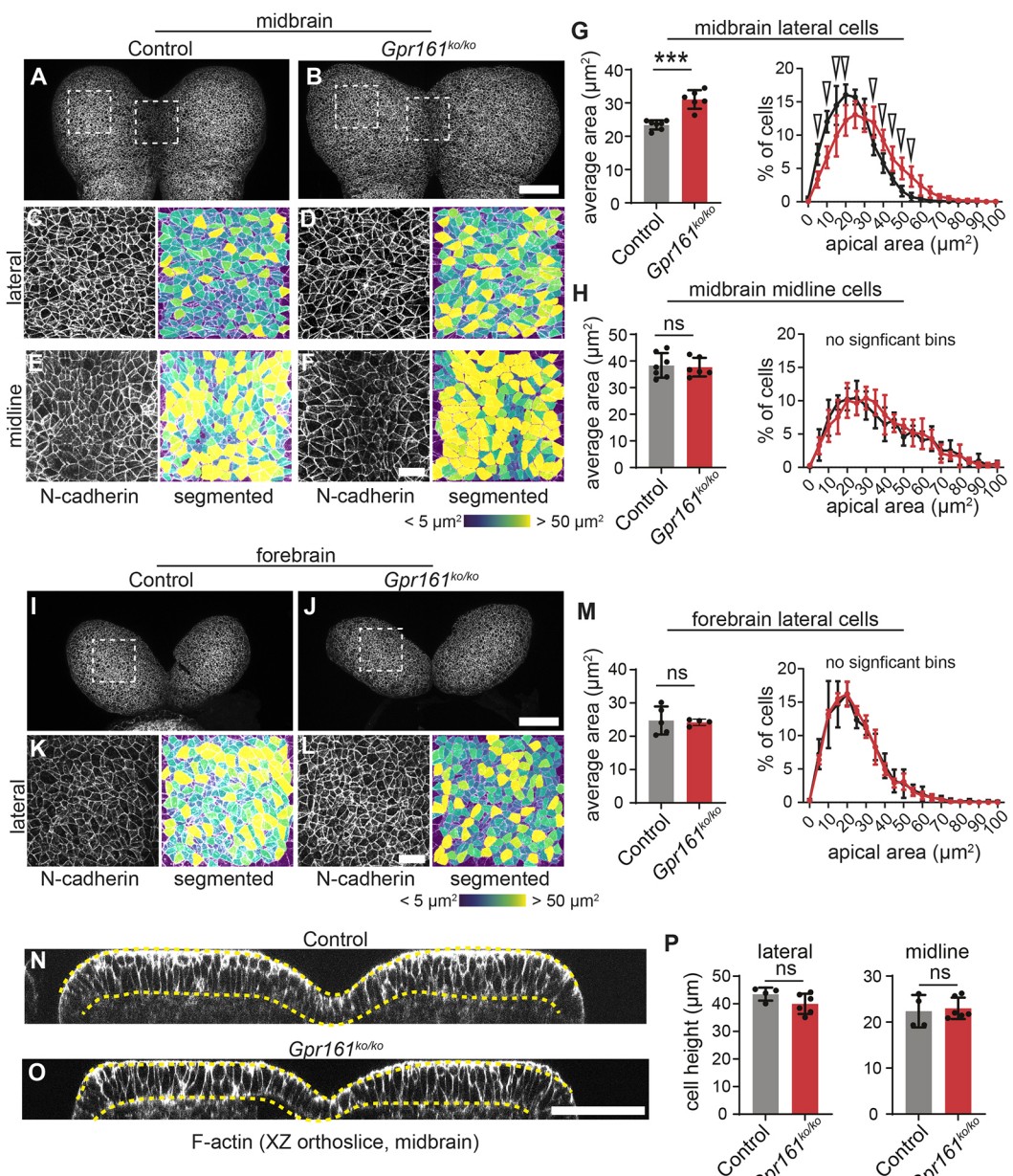

**Fig. 7. Loss of *Gpr161* leads to apical constriction defects in midbrain but not forebrain lateral cells.** (A,B) Tiled confocal images of the putative midbrain region of control and *Gpr161^{ko/ko}^* mutants at 6 somites. (C-F) Images of N-cadherin (cadherin 2) staining and resulting computational apical area segmentation from regions in the dashed boxes in A,B. Color in segmented images corresponds to apical area of individual cells. (G,H) Quantification of average apical area in midbrain cells from 5-7 somite stage embryos (left, each dot represents one embryo) and the proportion of cells displaying an apical area binned in 5 µm increments (right). Average lateral (control: 23.41±1.38 µm²; *Gpr161*: 31.06±2.80 µm²; *P*<0.001), but not midline (control: 38.34 ±4.63 µm²; *Gpr161*: 37.67±3.51 µm²; *P*=0.778), cell apical area is significantly increased in *Gpr161^{ko/ko}^*. Further, the bins designated by unfilled arrowheads in the lateral apical area distributions differ significantly (*P*<0.05 or lower) between control and *Gpr161^{ko/ko}^* mutants whereas no bins differed significantly in the midline distribution. (I,J) Tiled confocal images of the future forebrain region of the same control and *Gpr161^{ko/ko}^* mutant embryos. (K,L) N-cadherin staining and resulting computational segmentation are shown for the apical areas highlighted by the boxes. (M) Quantification of average apical area by embryo (left) and the distribution of areas (right). There is no significant difference in average apical area of forebrain cells between control and *Gpr161* mutants (control: 24.75±4.19 µm²; *Gpr161*: 24.32±0.95 µm²; *P*=0.848) and no bins significantly differ in the distribution analysis. (N,O) Single orthogonal views (*xz* reconstructions) at the level of the midbrain in control (N) and *Gpr161^{ko/ko}^* mutants (O). Dashed yellow lines indicate the apical (top) and basal (bottom) surfaces of the tissue. (P) Quantification of cell heights in the lateral and midline domains. Each dot represents one embryo. Note that no significant difference is observed at either position between control and mutant embryos. All data shown as mean±s.d. Statistical tests were unpaired two-tailed Student's *t*-test for average areas or height (****P*<0.001) and two-way ANOVA with Šidák's correction for area distributions. ns, not significant (*P*>0.05). Scale bars: 100 µm (A,B,I,J,N,O); 20 µm (C-F,K,L).

(Hwang et al., 2021) and a total lack of both activity pools in *Gpr161* ko prevents closure of both anterior and midbrain-hindbrain neuropores, but the remaining extraciliary activity of GPR161 in *Gpr161^{mut1/mut1}^* can drive initial cranial neural tube closure, but not fully establish the tissue-level changes required to maintain closure. Such differential cranial phenotypic outcomes in anterior versus midbrain-hindbrain neuropore closure and maintenance could be also relevant for other region-specific pathologies, such as encephaloceles (David, 1993).

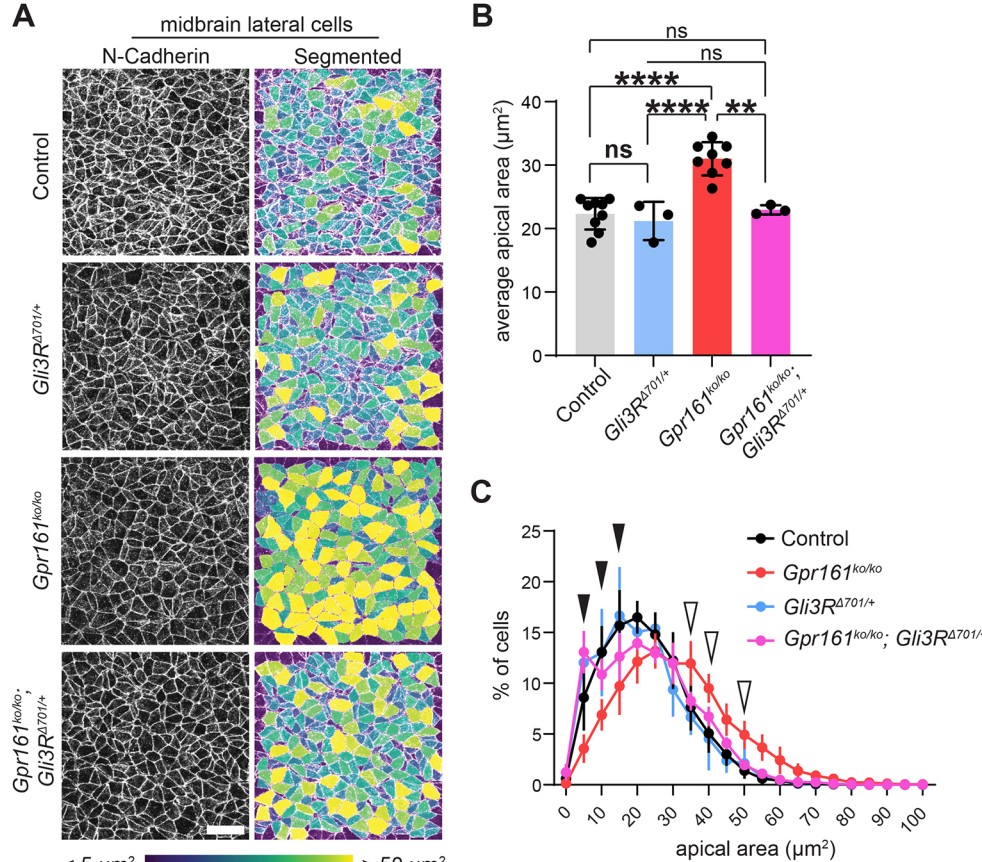

**Fig. 8.** *Gli3R* **expression rescues midbrain apical constriction defects in** *Gpr161* **mutant embryos.** (A) Images of N-cadherin staining and resulting computational segmentation from 100 µm² regions in the lateral midbrain region in control, *Gli3R^Δ701/+^*, *Gpr161^ko/ko^* and *Gpr161^ko/ko^; Gli3R^Δ701/+^* embryos at 5-7 somites. Color in segmented images corresponds to apical area of individual cells. (B) Plot of the average apical area of cells in each genotype, with each dot representing one embryo. *Gli3R* expression alone demonstrates no significant differences to control and expression of *Gli3R* in *Gpr161^ko/ko^* mutants rescues apical areas to control levels. Statistical differences determined by ordinary one-way ANOVA test with Tukey's correction. (**$P<0.01$, ****$P<0.0001$). ns, not significant ($P>0.05$). (C) Plot showing the proportion of cells displaying given apical areas (binned in 5 µm increments) for each genotype. Black arrowheads indicate bins for which *Gpr161^ko/ko^; Gli3R^Δ701/+^* embryos show significantly higher proportions of cells compared to *Gpr161^ko/ko^* alone; unfilled arrowheads indicate where they show significantly lower proportions of cells. Significance ($P<0.05$) was determined by a two-way ANOVA with Šidák's correction. A full set of pair-wise comparisons of distributions is shown in Fig. S4. All data shown as mean±s.d. Scale bar: 20 µm.

## HH mediated dorsoventral patterning is directly regulated by GLI3R in forebrain

HH target activation in control embryos is limited to the midline and juxta-midline domains, while active target repression occurs dorsolaterally (Fig. 9C) (Persson et al., 2002; Hwang et al., 2023). Our results show that FOXA2 forebrain expansion in *Gpr161* ko is rescued by expressing *Gli3R* but not by *Gli2* ko background. In contrast, FOXA2 hindbrain expansion in *Gpr161* ko is rescued by either *Gli3R* or *Gli2* ko background. However, absence of the PAX6 dorsal domain in *Gpr161* ko is restored either by expressing *Gli3R* or by *Gli2* ko in both the forebrain and hindbrain. Thus, *Gli3R* expression but not *Gli2* loss has a selective restorative effect on expansion of forebrain FOXA2 (Fig. 5D). We also noted high *Ptch1* levels throughout the cranial folds of *Gpr161* ko embryos compared to controls prior to axial rotation. Importantly, *Gli3R* expression restricted high HH signaling in the elevating cranial neural folds of *Gpr161* ko embryos prior to closure. Given that high *Ptch1* levels were not seen in younger *Gpr161* ko embryos, our results suggest that the lack of cranial neural fold apical constriction in the midbrain lateral folds occurs simultaneously with or preceding high HH-induced cell fate ventralization. However, some mutants with defective GLI3R generation, including *Ankmy2* ko, display de-repression of HH signaling as early as E7.75 (Somatilaka et al., 2020). Intriguingly, while *Gli3R* expression delays embryonic lethality in *Ankmy2* ko embryos, it does not rescue exencephaly and only partially reverses cranial neural tube ventralization (Hwang et al., 2023). Together, our data suggest that there may be additional regulatory layers that act to limit HH transduction before the full completion of neural induction, and this more global repression program could be crucial for promoting the morphogenetic changes driving cranial closure.

## Gli3R-regulated apical remodeling drives cranial closure

Our analysis of the role of specific downstream GLI effectors in cell remodeling yielded significant insights into the apical constriction program that drives the dramatic tissue curvature changes of cranial closure (Fig. 9). First, we can now conclude that closure defects observed in mutants with excessive and ectopic HH signaling do not result from overexpression or loss of expression in GLI2-dependent targets. These results extend our prior observations that loss of *Gli2* alone did not prevent apical constriction of lateral cells (Brooks et al., 2020), and are in line with the fact that *Gli2* mutants do not exhibit cranial closure defects (Mo et al., 1997; Matise et al., 1998; Bai and Joyner, 2001). Further, the observation that cranial closure in *Gpr161* mutants was rescued by expression of a constitutively repressive variant of GLI3 indicates that the apical constriction defects observed in excessive/ectopic HH signaling mutants arise from a failure to appropriately repress the expression of one or more negative targets of the pathway. Identifying these targets and determining how they modulate apical constriction will be an important next step in unraveling the spatial signaling logic that encodes this crucial morphogenetic behavior in cranial tissues.

Strikingly, excessive HH signaling only blocked apical constriction in the presumptive midbrain region of the cranial tissues, but forebrain cell remodeling was equivalent between control and *Gpr161* ko embryos, indicating that apical constriction blockade specifically in the midbrain is sufficient to block closure broadly across the cranial region. How does this local defect in cell remodeling result in such long-range consequences? One hypothesis is that coordinated apical constriction between the forebrain and midbrain regions is required to overcome the

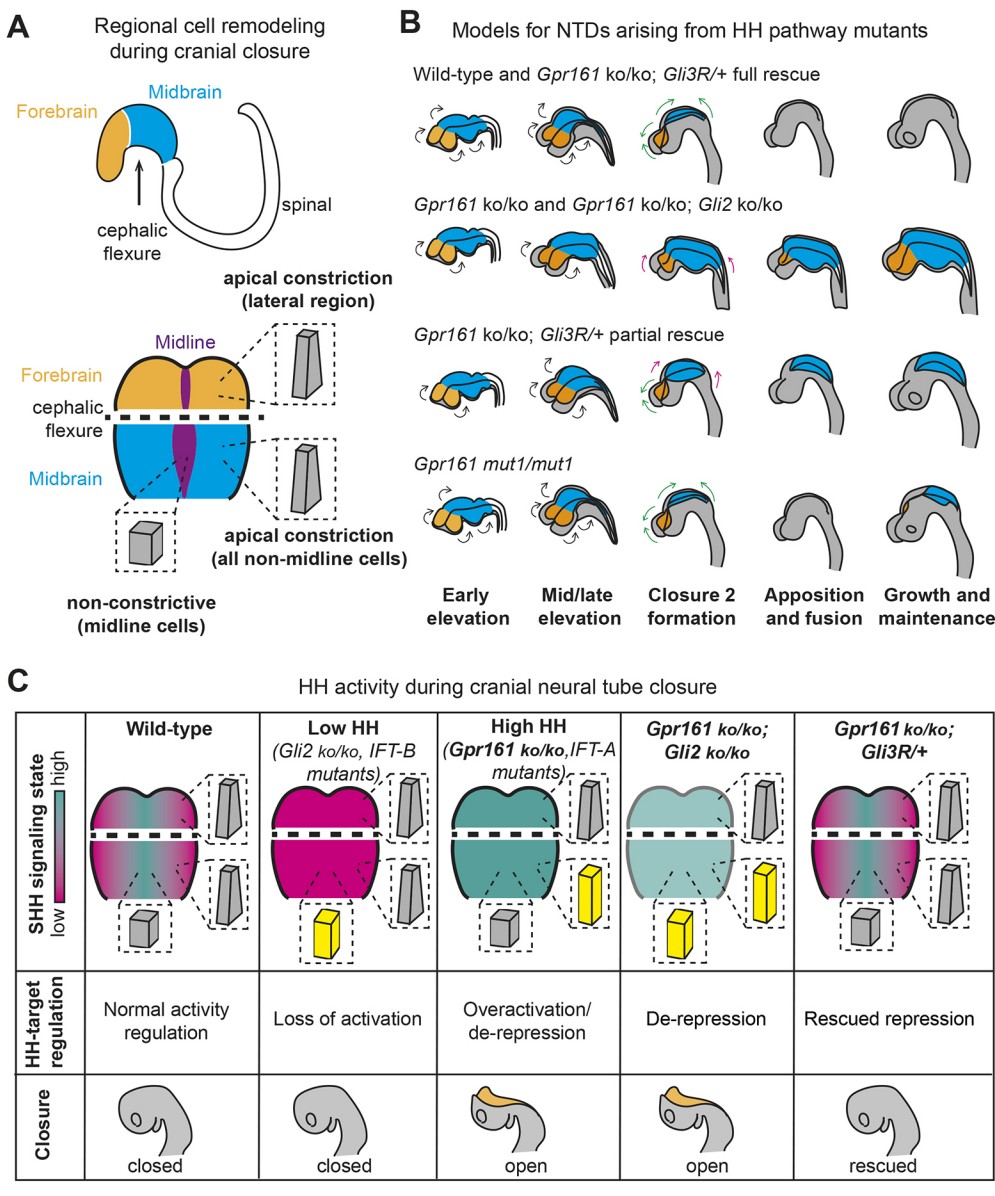

**Fig. 9. Summary of the role of GPR161 and the GLI factors in ciliary signaling and cell remodeling during cranial neural tube closure.** (A) A side-on schematic view of a mouse embryo at neural fold elevation stage, highlighting the prospective forebrain (orange) and midbrain (blue) tissues as well as the position of the cephalic flexure (top). Schematic summarizing the regional cell remodeling programs of cranial neural tube closure (bottom). Prospective lateral forebrain and midbrain tissues undergo apical constriction, whereas midline cells in the midbrain adopt a short and apically expanded architecture. (B) Schematics of the onset and progression of cranial closure defects. During wild-type and fully rescued closure in *Gpr161^{ko/ko}; Gli3R^{Δ701/+}*, robust apical constriction in the forebrain and midbrain act to bring the lateral edges of the neural tissues up above the midline so that they can undergo apposition and the zippering activity of fusion. Closure is then maintained as the volume of cranial tissues increases. In *Gpr161^{ko/ko}*, *Gpr161^{ko/mut1}* and *Gpr161^{ko/ko}; Gli2^{ko/ko}* embryos, the loss of apical constriction in the midbrain and robust elevation of this region are lost. While elevation still occurs in the forebrain, it alone is not sufficient to overcome the impediment raised by the cephalic flexure leaving almost all of the cranial region open from stalled apposition and fusion. In a small number of *Gpr161^{ko/ko}; Gli3R^{Δ701/+}*, elevation of the midbrain is only partially rescued and is therefore not sufficient to restore full apposition and fusion, resulting in closure of the forebrain but stalling in the mid- and hindbrain regions. Finally, in *Gpr161^{mut1/mut1}* embryos lacking GPR161-mediated ciliary signaling, the early stages of cranial closure appear to proceed largely normally in most embryos. However, these embryos appear to be unable to maintain closure once the volume of cranial tissues rapidly increase. Black arrows mark elevation forces, whereas green and magenta arrows indicate functional or nonfunctional apposition/fusion activity, respectively. (C) Summary of the consequences of various perturbations to HH signaling on the patterned cell remodeling programs and the progression of cranial closure. In wild-type embryos, HH activity is high at the midline and low laterally leading to normal cell remodeling. In mutants in which HH signaling is lower, midline cells become inappropriately tall, but lateral cells apically constrict as normal and cranial closure completes normally. In mutants that lead to high levels of HH signaling globally, such as *Gpr161^{ko/ko}*, enlarged apical cell areas are seen in lateral mid brain, but midline cell remodeling in midbrain is normal, as are apical cell areas in the forebrain region. If the downstream activator *Gli2* is deleted in a *Gpr161^{ko/ko}* background, midbrain apical constriction still fails. However, if the repressive activity of GLI3 is restored, lateral apical constriction in the midbrain is restored.

significant mechanical impediment represented by the sharp curvature of the cephalic flexure, a region of significant bending in the putative midbrain region (Fig. 9) (Jacobson and Tam, 1982), particularly as there is a correlation between curvature and the

observed rate of closure progression in more posterior tissues (Peeters et al., 1998). In this model, the loss of midbrain elevation in *Gpr161^{ko/ko}* embryos leaves the neural folds widely spaced across this flexure (Fig. 9B), stalling the already slow progression of the

zippering activity. Subsequent growth of the tissue then largely undoes any anterior closure accomplished by forebrain apical constriction Alternatively, the disjoint in elevation between the forebrain and midbrain regions could result in a failure to establish closure 2 (Yamaguchi and Miura, 2013; Juriloff and Harris, 2018), leading to widespread forebrain and midbrain exencephaly.

Our data also suggest that apical constriction in the forebrain and midbrain are under differential genetic control (Fig. 9C). It remains to be seen whether this difference is only in the initiating signals or is also found in the downstream machinery physically powering cell remodeling. Intriguingly, apical constriction defects in the midbrain are associated with errors in actomyosin organization (Brooks et al., 2020; Bogart and Brooks, 2025), whereas cells in the mammalian forebrain depend on membrane remodeling by the endocytic receptor LRP2 for constriction and closure (Kur et al., 2014; Sabatino et al., 2017; Kowalczyk et al., 2021). However, it remains unclear whether these are regionally unique apical constriction mechanisms, or if a combination of endocytosis and contractility are required in both populations. In either case, the sufficiency of GPR161-dependent midbrain apical constriction in cranial closure suggests that high GLI3R thresholds predominantly support midbrain-specific features or sensitivity in the morphogenetic programs powering cranial closure.

### Limitations of the current study

One consistent theme that emerged in this study was the spatial complexity of HH signal transduction. Patterns in both cell fate and cell shape remodeling showed strikingly different sensitivity to manipulation of GLI2 and GLI3 activity depending on where the cells were along the anterior-posterior axis. This builds upon our recent work demonstrating that forced activation of HH signaling has metamere-specific consequences for gene expression in different regions of the cranial tissues (Brooks et al., 2025), and suggests that the separation between the roles of the activating and repressive arms of the HH pathway could be crucial factors that integrate the mediolateral and dorsoventral patterning systems. This complexity also prevented us from identifying the specific target(s) of HH signaling that modulates apical constriction, as such an investigation would require the application of targeted spatial multi-omics in a variety of complex genetic backgrounds. In the future, identification and further exploration of such spatiotemporally regulated targets of activation and repression in this tissue will allow us, not only to understand how HH signals mediate spatially restricted cell remodeling programs, but also how such dynamic cell behaviors are integrated with the cell fate trajectories influenced by local HH activity states. Such studies will also provide fundamental groundwork that could eventually lead to targeted genetic interventions aimed at ameliorating cranial defects arising from dysregulation of the HH pathway in humans.

### MATERIALS AND METHODS

#### Mouse strains

All mice were housed at the Animal Resource Center of the University of Texas Southwestern (UTSW) Medical Center. All the animals in the study were handled according to protocols approved by the UT Southwestern Institutional Animal Care and Use Committee, and the mouse colonies were maintained in a barrier facility at UT Southwestern, in agreement with the State of Texas legal and ethical standards of animal care. Mice were housed in standard cages that contained three to five mice per cage, with water and standard diet *ad libitum* and a 12 h light/dark cycle. Both male and female mice were analyzed in all experiments. Noon of the day on which a vaginal plug was found was considered E0.5. Staging of embryos was further refined by counting somites. The *Gpr161* knockout and conditional allele targeting

the third exon crossed has been described before (Hwang et al., 2018). The *Gpr161* knockout and conditional lines are already indexed in MGI as *Gpr161$^{tm1.1Smuk}$* (MGI: 6357708) and *Gpr161$^{tm1.2Smuk}$* (MGI: 6357710). The *Gpr161$^{mut1}$* allele has been described before (Hwang et al., 2021) and is indexed in MGI as *Gpr161$^{tm2.1Smuk}$* (MGI: 6763974). Double knockout analysis was performed using *Gli2$^{tm1Alj}$* (ko) allele (Mo et al., 1997). The *Gli2* ko and *Gpr161* floxed alleles were linked through genetic recombination by breeding *Gpr161$^{f/f}$* with *Gli2$^{ko/+}$* animals. Heterozygotes of the ubiquitously recombined *Gli3$^{\Delta701C}$* allele are embryonic lethal (Cao et al., 2013), suggesting that the *Gli3$^{\Delta701}$* allele produces a much more potent form of GLI3R than the widely used *Gli3$^{\Delta699}$* allele, which yields viable heterozygotes (Bose et al., 2002). Therefore, timed pregnancies were set up between *Gli3$^{\Delta701C/\Delta701C}$; Gpr161$^{ko/+}$* or *Gli3$^{\Delta701C/\Delta701C}$; Gpr161$^{ko/f}$* and *CAG-Cre; Gpr161$^{ko/+}$* breeders for generating embryos expressing *Gli3R$^{\Delta701}$* and lacking *Gpr161. CAG-Cre* is expressed ubiquitously (Sakai and Miyazaki, 1997). Yolk sac DNA was used for genotyping embryos. Embryonic stem cells for *Gli3$^{\Delta701C}$* (Cao et al., 2013) were injected by the UT Southwestern transgenic core into C57BL/6N blastocysts and selected for germline transmission among chimeras and backcrossed into C57BL/6J background.

#### Mouse genotyping

Genotyping of *Gpr161$^{mut1}$* alleles was performed using primers in introns 3-4 (5′-CAGAAAGCAACAGCAAAGCA-3′) and 4-5 (5′-ACCCTGA-CACTGCCCTTAGC-3′). The PCR product of wild-type and *mut1* allele bands was 927 bp, but only the PCR product from the *mut1* allele was digested into 400 and 527 bp products with NotI. Genotyping of *Gpr161* knockout or floxed alleles were performed using primers in the deleted 4th exon (5′-CAAGATGGATTCGCAGTAGCTTGG-3′), flanking the 3′ end of the deleted exon (5′-ATGGGGTACACCATTGGATACAGG-3′), and in the Neo cassette (5′-CAACGGGTTCTTCTGTTAGTCC-3′). Wild-type, floxed and knockout bands were 816, 965 and 485 bp, respectively (Hwang et al., 2018). The *Cre* allele was genotyped with Cre-F (5′-AATGCTGTC ACTTGGTCGTGGC-3′) and Cre-R (5′-GAAAATGCTTCTGTCCGTT-TGC-3′) primers (100 bp amplicon). To genotype *Gli2* mice, Gli2 sense (5′-AAACAAAGCTCCTGTACACG-3′), Gli2 antisense (5′-CACCCCAAAG-CATGTGTTTT-3′) and pPNT (5′-ATGCCTGCTCTTTACTGAAG-3′) primers were used. Wild-type and knockout bands were 300 bp and 600 bp, respectively. Genotyping of the *Gli3$^{\Delta701C}$* allele was carried out with the following primers: BW892F (5′-AATG-GAATGTTTCCAAGA-CTG-3′) and BW892R (5′-ATAAAACCAAGGGTTCCAGATC-3′), with wild-type and mutant bands being 180 bp and 250 bp, respectively (Cao et al., 2013). *Cre*-mediated recombination was confirmed using BW791 (5′-GACCTCATCTTTAGCTTTGCC-3′) and BW1020R (5′-CAAGGGTTC-CAGATCTGGATC-3′), the recombined allele band being 230 bp.

#### Tissue processing, immunostaining and microscopy

Mouse embryos fixed in 4% paraformaldehyde overnight at 4°C and processed for cryosectioning. For cryosectioning, the embryos were incubated in 30% sucrose at 4°C until they were submerged in the solution. Embryos were then mounted with OCT compound and cut into 15-μm-thick frozen sections. The sections were incubated in PBS for 15 min to dissolve away the OCT. Sections were then blocked using blocking buffer [1% normal donkey serum (Jackson ImmunoResearch) in PBS] for 1 h at room temperature. Sections were incubated with primary antibodies against the following antigens; overnight at 4°C: FOXA2 (1:1000; ab108422, Abcam; RRID: AB_11157157), NKX2-2 (1:100; 74.5A5-s, DSHB; RRID: AB_531794), OLIG2 (1:500; MABN50, Millipore; RRID: AB_10807410), NKX6-1 (1:100; F55A10-s, DSHB; RRID: AB_532378), PAX6 (1:2000; 901301, BioLegend; RRID: AB_2749901). After three PBS washes, the sections were incubated in secondary antibodies (Alexa Fluor 488- and 594-conjugated secondary antibodies; 1:500; 711-585-152, 715-545-151, 715-585-150, Jackson ImmunoResearch) for 1 h at room temperature. Cell nuclei were stained with DAPI. Slides were mounted with Fluoromount-G (0100-01, Southern Biotech) and images were acquired with a Zeiss AxioImager.Z1 microscope. For whole mounts, after fixation embryos were washed three times for 30 min in PBS+0.1% Triton X-100 (PBT; Fisher Scientific) and then blocked for 1 h in a blocking solution consisting of PBS+10% bovine serum albumin (Thermo Fisher Scientific) for 1 h at room

temperature. Embryos were then incubated with primary antibodies (rabbit anti-N-Cadherin; 1:500; Cell Signaling Technology, 13116; RRID: AB_2687616) overnight at 4°C. Embryos were then washed three times for 30 min in PBT, incubated in secondary antibodies (anti-rabbit Alexa Fluor 488, A-11008, 1:500, Thermo Fisher Scientific) for 1 h at room temperature, and washed three times for 30 min in PBT. After staining, embryos were mounted dorsal side down against #1.5 coverglass in PBT in Attofluor chambers (Thermo Fisher Scientific, A7816) using a small fragment of broken cover glass with dabs of vacuum grease to gently hold the embryo against the glass. Embryos were imaged on a Zeiss LSM 980 confocal microscope equipped with a Plan-NeoFluar 40×/1.3 oil immersion objective. Images were captured using tile-based imaging to acquire contiguous z-stacks of 50-100 μm depth, with an optical slice thickness of 0.6 μm and z-steps of 0.3 μm. Tile stitching and maximum intensity projection were performed in Zeiss ZEN Blue software.

## SEM
Embryos were fixed in ½ Karnovsky's fixative (2% paraformaldehyde, 2.5% glutaraldehyde in 0.1 M sodium cacodylate buffer, pH 7.4) overnight and post-fixed in 1% $OsO_4$ for 1 h. Embryos were dehydrated through a series of ethanol. After three washes of hexamethyldisilazane, the samples were air-dried at room temperature. The embryos were oriented and mounted on carbon tape on aluminum stubs. They were then sputter-coated with 10 nm of gold/palladium mixture and viewed on an FEI XL30 scanning electron microscope at 10 kV.

## ISH
Antisense riboprobes were made using the mouse *Ptch1* template (from Andrew McMahon lab, CalTech, CA, USA). Whole-mount ISH using digoxigenin-labeled probes was performed on embryos using standard protocols (Hwang et al., 2018). Images were acquired using a Leica stereomicroscope (M165 C) with digital camera (DFC500).

## Image analysis and quantification
All quantification was performed on unprocessed images. Apical cell area was quantified in 100 μm×100 μm regions of the tissue at various positions from embryos at 5-7 somites. These including the midbrain lateral domain, defined here as approximately halfway between the midline and lateral edge of the tissue and halfway between the pre-otic sulcus and the cranial flexure; the midbrain midline; and the forebrain lateral domain, defined as halfway between the anterior neural ridge and the cranial flexure and halfway between the midline and lateral edge of the tissue. Cells entirely contained within these regions were segmented using CellPose (Stringer et al., 2021) and cell areas were quantified and displayed using the MorphoLibJ plugin (Legland et al., 2016) in Fiji/ImageJ (Schindelin et al., 2012; Schneider et al., 2012). Tissue thickness was measured in orthogonally reconstructed z-stacks of midbrain tissues, midway between the midline and lateral edge of the tissue. Quantification of FOXA2 and PAX6 expression in that of the neural tube was done from horizontal sections at forebrain and hindbrain regions utilizing the line tool in ImageJ (Hwang et al., 2021, 2023).

## Statistical analysis
For apical constriction, statistical analyses included unpaired Student's t-test for comparing two genotypes (Fig. 7G,H,M,P), ordinary one-way ANOVA with Tukey's correction for comparing four genotypes (Fig. 8B), and two-way ANOVA with Šidák's correction for multiple comparisons for comparing two or more cell area distributions (Fig. 7G,H,M, Fig. 8C, Fig. S4). Summary statistics were as follows: *$P$<0.05; **$P$<0.01; ***$P$<0.001; ****$P$<0.0001. Statistical testing and graph generation were performed in Prism (GraphPad) and figures were assembled using Photoshop and Illustrator (Adobe).

## Acknowledgements
We thank the transgenic core for generating *Gpr161mut1*, molecular pathology core for histology support, and mouse animal care facilities for animal care in UT Southwestern. We thank Phoebe Doss for SEM support, and anonymous reviewers and S.M. lab members for comments on the manuscript.

## Competing interests
The authors declare no competing or financial interests.

## Author contributions
Conceptualization: E.R.B., S.-H.H., S.M.; Data curation: E.R.B., S.-H.H.; Formal analysis: E.R.B., S.-H.H., S.M.; Funding acquisition: E.R.B., S.M.; Investigation: E.R.B., S.-H.H., K.A.W.; Methodology: E.R.B., S.-H.H., K.A.W.; Project administration: E.R.B., S.M.; Resources: E.R.B., S.-H.H., S.M.; Supervision: E.R.B., S.M.; Validation: E.R.B., S.-H.H., K.A.W.; Visualization: E.R.B., S.-H.H.; Writing – original draft: E.R.B., S.M.; Writing – review & editing: E.R.B., S.-H.H., S.M.

## Funding
This study was supported by the National Institute of General Medical Sciences (1R35GM144136 to S.M.) and startup funds from North Carolina State University (E.R.B.). The content is solely the responsibility of the authors and does not necessarily represent the official views of the National Institutes of Health. The funders had no role in study design, data collection and analysis, decision to publish, or preparation of the manuscript. Open Access funding provided by University of Texas Southwestern Medical Center. Deposited in PMC for immediate release.

## Data and resource availability
All relevant data and details of resources can be found within the article and its supplementary information.

## The people behind the papers
This article has an associated 'The people behind the papers' interview with some of the authors.

## Peer review history
The peer review history is available online at https://journals.biologists.com/dev/lookup/doi/10.1242/dev.205171.reviewer-comments.pdf

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
