## [Peer Review File · Development (Cambridge, England)]

GPR161-GLI3 repressor signaling at cilia directs apical constriction and cell fate during cranial neural tube closure

Eric R. Brooks, Sun-Hee Hwang, Kevin A. White and Saikat Mukhopadhyay

DOI: 10.1242/dev.205171

Editor: James Briscoe

Review timeline

Original submission:	16 August 2025
Editorial decision:	8 September 2025
First revision received:	30 October 2025
Editorial decision:	5 November 2025
Second revision received:	6 November 2025
Accepted:	7 November 2025

Original submission

First decision letter

MS ID#: dev.205171

MS Title: GPR161-GLI3 repressor signaling at cilia directs apical constriction and cell fate during cranial neural tube closure

Authors: Eric R. Brooks; Sun-Hee Hwang; Kevin A. White; Saikat Mukhopadhyay

Article Type: Research Article

Dear Dr Mukhopadhyay,

I have now received all the referees' reports on the above manuscript, and have reached a decision. The referees' comments are appended below, or you can access them online: please go to: View Reviewer Comments

As you will see, the referees express considerable interest in your work, but have some significant criticisms and recommend a substantial revision of your manuscript before we can consider publication. Please clarify the neural tube closure phenotypes and include data on closure timing and proportions of embryos with open neuropores. An important issue is addressing how midbrain-specific apical constriction defects translate into forebrain exencephaly phenotypes. Additionally, please provide clearer statistical analyses, verify that Gli2 knockout reduces HH signaling in your experimental context, and correct gene nomenclature inconsistencies.

If you are able to revise the manuscript along the lines suggested, which may involve further experiments, I will be happy to receive a revised version of the manuscript. Your revised paper will be re-reviewed by one or more of the original referees, and acceptance of your manuscript will depend on your addressing satisfactorily the reviewers' major concerns. Please also note that Development will normally permit only one round of major revision. If it would be helpful, you are welcome to contact us to discuss your revision in greater detail. Please send us a point-by-point response indicating your plans for addressing the referees' comments, and we will look over this and provide further guidance.

Please attend to all of the reviewers' comments and ensure that you upload both a 'clean' version of your Word file, along with a highlighted version clearly showing where you have made changes in the revised manuscript. Please avoid using 'Tracked changes' in Word files as these are lost in PDF conversion. I should be grateful if you would also provide a point-by-point response detailing how you have dealt with the points raised by the reviewers in the 'Response to Reviewers' box. If you do not agree with any of their criticisms or suggestions please explain clearly why this is so.

Reviewer 1

SUMMARY OF THE ADVANCE MADE IN THIS PAPER AND ITS POTENTIAL SIGNIFICANCE TO THE FIELD

This manuscript by Brooks et al. investigates the role of the G protein-coupled receptor GPR161 and GLI effectors in cranial neural tube closure. Using an allelic series, epistasis experiments, scanning electron microscopy, and immunofluorescence, the authors show that extraciliary GPR161 can partially compensate for the loss of GPR161 to delay exencephaly, and that exencephaly and forebrain patterning defects in Gpr161 knockout embryos can be rescued by constitutive Gli3R expression but not by loss of Gli2 (in contrast to the spinal cord and hindbrain). These findings are interesting for several reasons. First, the region-specific requirements highlight important differences between forebrain, hindbrain, and spinal cord closure mechanisms. Second, the data suggest that derepression (rather than excessive activation) of Hedgehog signaling underlies cranial closure defects in Gpr161 mutants, consistent with known effects of Gpr161 loss on GLI3R and ciliary GLI2. Third, building on prior work, the authors show that GLI3R expression rescues midbrain lateral cell apical constriction defects, leading to rescue of the anterior neuropore closure and the formation of optic cups in Gpr161 mutants.

The manuscript provides strong genetic and imaging evidence that GLI3 repressor is key for mediating cranial closure downstream of GPR161. Demonstrating that constitutive GLI3R expression can rescue exencephaly is a useful addition to our understanding of Hedgehog pathway regulation and could provide interesting insights into the underlying mechanisms that drive exencephaly. However, in my view the conceptual advance is somewhat limited. Respectfully, prior work from both Dr. Brooks (Brooks et al. 2020; Brooks et al. 2025) and Dr. Mukhopadhyay (Mukhopadhyay et al. 2013; Hwang et al. 2021) has already established that excessive Hedgehog signaling impairs apical constriction, that GPR161 promotes GLI3R formation, and that cranial closure involves metamere-specific logic. This manuscript consolidates these ideas, but does not introduce a fundamentally new paradigm. The novelty lies in demonstrating the sufficiency of GLI3R to rescue the anterior neuropore and optic cup phenotypes in the absence of functional GPR161, but this represents an incremental rather than transformative advance. For this reason, I am not convinced the work meets the high bar of conceptual advance and broad interest required for publication in Development.

SUGGESTIONS TO AUTHORS

-Limitations of the study that constrain impact: The study lacks mechanistic insight linking Gpr161 loss or GLI3R addition to apical constriction. Is this direct? Could it be due to changes in cell type/identity? To strengthen this paper, please clarify why this particular region of the brain (containing the midbrain lateral cells) is so sensitive to Gpr161 loss or how the apical area can be rescued with the addition of GLI3R. More specifically, as alluded to in the discussion, identify the direct GLI3R targets in the midbrain or try to define the mechanisms that underlie the apical restriction defects.

-Clarification: The manuscript convincingly shows that GLI3R expression rescues apical constriction defects in the midbrain lateral cells. However, the overt exencephaly phenotype is described primarily in the forebrain (failure of anterior neuropore closure and absence of optic cups). As written, it was unclear to me how a midbrain-specific defect translates into a more anterior phenotype. Perhaps, the authors could explicitly articulate this connection (as depicted in Figure 9) more clearly, more specifically how midbrain apical constriction is the critical mechanical driver of cranial fold elevation and that its failure prevents anterior neuropore closure.

-Questions about midbrain specificity: Perhaps I misunderstood the paper, but it seems to me that both the presence of the apical constriction defects in the Gpr161 mutants and the effects of the GLI3R expression rescue are specific to very limited regions of the developing midbrain. Can the authors speculate more as to why this might be?

-Minor issue: The gene name is inconsistently written as "Grp161" instead of "Gpr161" in several places and should be corrected.

Reviewer 2

SUMMARY OF THE ADVANCE MADE IN THIS PAPER AND ITS POTENTIAL SIGNIFICANCE TO THE FIELD

This excellent new paper from extends the Brooks Lab's exciting recent work on the interplay of cilia-mediated HH signaling and the cell biology of neural tube closure in mammals, and in this new collaboration with the more signaling-focused Mukhopadhyay Lab, the paper represents what I hope will be the first of many more such studies. It's a perfect fit for Development, and I strongly recommend publication with only minor comments.

The paper extends previous work exploring how dysregulation of cilia-mediated HH signaling evokes neural tube closure defects (NTDs) in the mouse. Here, they have performed a careful study of NTDs and signaling in mice with mutations in the HH transducer Gpr161, which notably is a known genetic risk factor for human NTDs. Using SEM and whole mount staining and sections, they explore both the knockout and also a knockin mutant that renders Gpr161 unable to enter and cilia and signal via cAMP. They follow this with an equally careful study of the genetic interaction between these alleles and mutant in the Gli2 activator and the Gli3 repressor. They also explore the A/P and D/V patterning of the brains in all of these mice, finding quite specific patterns of disruption. Here the outcome is very clear: de-repression, not excessive activation, of HH targets cause the NTDs. This work alone, involving some onerous genetics and careful analysis of important proteins with clinical implications would make the paper worthy of Development. But the second half of the paper is even cooler.

Shifting from genetic to cell biology, they use quantitative imaging to analyze the cell behaviors underlying these NTDs and find that these mice diverge in subtle but important ways from previously examined HH-related mutants. Notably, while the entire cranial neural tube fails to close, defects in apical constriction were observed only in the midbrain and not the forebrain. These findings are especially important because human NTDs are wildly heterogeneous disorders, and in these experiment the authors begin to model that complexity in mouse models.

This represents a lot of work, and the data are to me very convincing. I urge the Editor to resist any calls by other reviewers asking for the intangible "more."

SUGGESTIONS TO AUTHORS

Minor Comments:

1. It will be useful, I think, to be more explicit about the stats performed to compare histograms, in the text and the main figure. I was chiding the authors mentally until I finally noticed the little arrowheads in Fig 8 and Supp. Fig. 8. And was such an analysis done for Fig 7G, h, M? The differences seem wildly significant, but no arrowheads there...
2. I think reviewers would benefit if the summary model in Fig. 9 included other HH pathway mutants from previous papers (Gli or Ift mutants for example). This will help the uninitiated reader better grasp the complexity of genotype/phenotype in this interesting series of NTD mutants.
3. Along the same lines, for readers who may be expert in neural morphogenesis but less familiar with HH signaling and cilia, could a diagram be added that sketches the logic of HH/Gpr161/Gli signaling? This would especially be useful for distinguishing Gli2 and Gli3 effects.

Reviewer 3

In this substantial manuscript, Brooks et al use allelic series to disentangle the contributions of SHH signalling through activation versus repression of genes to regulation of neurulation and anterior

neural patterning in mice. The paper is packed with fascinating findings, such as regional differences in apical constriction regulation between the forebrain and midbrain and inter-relation between neurulation and eye development - with anophthalmia and exencephaly being commonly concurrent in mouse models. The starting point for the work is previous studies by these and other authors showing that loss of Gpr161 causes exencephaly, de-repression or over-activation of SHH signalling, and that anterior neuroepithelial cells with high SHH signalling do not apically constrict. The authors first compare the phenotype of Gpr161 loss of function with a variant unable to localise to the cilia. They then combine Gpr161 knockout alleles with loss of Gli2 or of Gli3R. Gli3R deletion rescued the neurulation phenotype whereas loss of Gli2 rescued neither neurulation nor over-expression of the SHH target Ptch1 in exencephalic neural tissue. The authors are asked to consider the following points in decreasing order of consequence:

1. "However, Gpr161mut1/mut1 embryos by E11 had externally closed cranial neural tube (Fig. 1C)... However, the Gpr161mut1/mut1 embryos developed exencephaly at later" and

"we note closure of the anterior neuropore despite an open midbrain-hindbrain neuropore at later embryonic stages."

The description of the neural tube phenotypes in the cilia-excluded Gpr161 model is not clear. Is this a model in which the anterior neural tube is fully closed (Closures 1-3 all form, HNP and ANP both close), or a model in which the HNP fails to close as suggested in the discussion? Failure of HNP closure is difficult to differentiate from other closure failures at late stages, when exencephaly is present. Do the authors have data on the proportion of embryos with open HNPs at different somite stages? A re-opening phenotype would be very exciting given the rarity of models. The authors may wish to discuss findings in McLaren et al Dec Cell 2025 which analyses post-closure roofplate mechanics.

2. Do any of the models analysed have grossly observable spinal phenotypes? If not, that is worth stating to avoid others killing mice to test them in future (and as further evidence of regionalisation of mechanisms).

3. Interpretation of the Gli2 KO allelic series depends on evidence that it achieved reduction in SHH signalling in the Gpr161-null background. They state "unlike the spinal cord, lack of Gli2 function was unable to restrict the excessive HH signaling promoted by Gpr161 loss in the open cranial neural tube". Spinal cord expression of Ptch1 is not described in the preceding text, but Figure 2C does appear to show lower staining in the spinal neural tube of Gpr161-Ko/Gli2-Ko than in Gpr161-ko alone. Is this reproducible? Exencephalic tissue can very 'sticky' to in situ probes and is exposed to abnormal signalling from the amniotic fluid (also seen in figure 4B?): might the spinal expression pattern more meaningfully represent HH status?

4. Rescue of the exencephaly phenotype by Gli3R Δ 701/+ is striking. While not critical to the interpretation of the data - do the authors know whether there is a sex bias in the rescue (given the well-documented female excess of exencephaly)? The methods state that both males and females were analysed so if that data is available, it would be worth including, although it probably does not meet the ethical bar for killing more mice to obtain in isolation.

5. What is more important for regulation of apical constriction needed for midbrain neural tube closure: the absolute level of HH signalling or its midline localisation? The E8.5 in situ shown suggest both change with the alleles used.

6. Figure 7H - the label above the horizontal line is mis-centred.

7. "Lack of Gli2 and Gli3R expression..." sounds like a double knockout model.

8. Line 189: "First, we first"

First revision

Author response to reviewers' comments

We thank the reviewers for their constructive comments and appreciate the very positive response

from the editors and reviewers (for e. g., Reviewer 2: “*excellent new paper*”, “*perfect fit for Development, and I strongly recommend publication with only minor comments*”; Reviewer 3: “*substantial manuscript*”, “*packed with fascinating findings*”; “*rescue of the exencephaly phenotype by *Gli3RΔ701/+* is striking*” and Reviewer 1 even with some reservation on conceptual advance: “*provides strong genetic and imaging evidence*”, “*findings are interesting for several reasons*”; “*useful addition to our understanding of Hedgehog pathway regulation*”).

We believe that we have now addressed most of the remaining concerns and revised the manuscript according to the suggestions. Notable changes in the revised submission include:

- o Extensive textual edits based on all three reviewers’ suggestions.
- o Revised Fig 1 including table and revised Suppl Figure 1 including additional data on characterization of closure defects in *Gpr161^{mut1/mut1}* embryos.
- o Revised Figures 2 and 5 with schematics showing GPR161’s role in HH pathway and cranial neural tube patterning
- o Revised Figures 7, 8
- o Revised Figure 9 updating summary model with GPR161 and other relevant mutants
- o Four additional figures are included in this rebuttal document to address reviewers’ comments.

All textual revisions in the revised submission are marked.

Reviewer 1: This manuscript by Brooks et al. investigates the role of the G protein-coupled receptor GPR161 and GLI effectors in cranial neural tube closure. Using an allelic series, epistasis experiments, scanning electron microscopy, and immunofluorescence, the authors show that extraciliary GPR161 can partially compensate for the loss of GPR161 to delay exencephaly, and that exencephaly and forebrain patterning defects in *Gpr161* knockout embryos can be rescued by constitutive *Gli3R* expression but not by loss of *Gli2* (in contrast to the spinal cord and hindbrain). These findings are interesting for several reasons. First, the region-specific requirements highlight important differences between forebrain, hindbrain, and spinal cord closure mechanisms. Second, the data suggest that derepression (rather than excessive activation) of Hedgehog signaling underlies cranial closure defects in *Gpr161* mutants, consistent with known effects of *Gpr161* loss on *GLI3R* and ciliary *GLI2*. Third, building on prior work, the authors show that *GLI3R* expression rescues midbrain lateral cell apical constriction defects, leading to rescue of the anterior neuropore closure and the formation of optic cups in *Gpr161* mutants. The manuscript provides strong genetic and imaging evidence that *GLI3* repressor is key for mediating cranial closure downstream of GPR161. Demonstrating that constitutive *GLI3R* expression can rescue exencephaly is a useful addition to our understanding of Hedgehog pathway regulation and could provide interesting insights into the underlying mechanisms that drive exencephaly. However, in my view the conceptual advance is somewhat limited. Respectfully, prior work from both Dr. Brooks (Brooks et al. 2020; Brooks et al. 2025) and Dr. Mukhopadhyay (Mukhopadhyay et al. 2013; Hwang et al. 2021) has already established that excessive Hedgehog signaling impairs apical constriction, that GPR161 promotes *GLI3R* formation, and that cranial closure involves metamere-specific logic. This manuscript consolidates these ideas, but does not introduce a fundamentally new paradigm. The novelty lies in demonstrating the sufficiency of *GLI3R* to rescue the anterior neuropore and optic cup phenotypes in the absence of functional GPR161, but this represents an incremental rather than transformative advance. For this reason, I am not convinced the work meets the high bar of conceptual advance and broad interest required for publication in *Development*.

We thank the reviewer for their positive comments regarding our experiments. Regarding the conceptual advance central to this paper, we feel that it goes beyond a simple consolidation of previous work. For example, while GPR161 promotes *GLI3R* production, lack of *Gpr161* results in both *GLI3R* loss and excessive *GLI2A* generation and both factors are associated with the consequences of excessive HH signaling. Indeed, although ventral neural progenitor expansion at all levels of the spinal neural tube (Hwang et al., 2023) and hindbrain (this manuscript) are rescued by either *Gli2* loss or *Gli3R* expression, the effects of these perturbations in cranial neural tube were unknown. The current manuscript demonstrates a specific role of *GLI3R* in both cranial closure and patterning that cannot be duplicated by reducing *GLI2A* activity. This is surprising and

novel finding and suggests a unique regulatory layer to how HH signals are interpreted along the anteroposterior axis. Given the multifarious roles of HH in many developmental contexts that rely on precise articulation of this activation/repression balance, we think many readers of *Development* will find this separability interesting and useful in their thinking about how this pathway controls spatially delimited cell fates and tissue morphogenesis.

SUGGESTIONS TO AUTHORS

-Limitations of the study that constrain impact: The study lacks mechanistic insight linking *Gpr161* loss or *GLI3R* addition to apical constriction. Is this direct? Could it be due to changes in cell type/identity? To strengthen this paper, please clarify why this particular region of the brain (containing the midbrain lateral cells) is so sensitive to *Gpr161* loss or how the apical area can be rescued with the addition of *GLI3R*. More specifically, as alluded to in the discussion, identify the direct *GLI3R* targets in the midbrain or try to define the mechanisms that underlie the apical restriction defects.

Thank you for this comment, and we have now made this point explicitly in the Limitations section of the Discussion in the revised manuscript. In our view, our data establishing a specific role of *GLI3R* in regulating exencephaly and our detailed analysis of how HH activity modulates both region-specific apical constriction and neural patterning during cranial neural tube closure represent a major conceptual and technical advance. Given the complexity of the regulatory logic we are unraveling, the next major advance will be to identify and functionally study forebrain, midbrain and hindbrain-specific targets of both *GLI2A* and *GLI3R* at stages ranging from E8-E10. While we have performed previous single cell transcriptomic analyses at this stage (Brooks et al., 2025) to identify HH responsive targets, determining targets that specifically respond to *GLI3R* or *GLI2A* and characterizing their roles in cranial neural tube closure and patterning will require additional targeted sequencing and likely a more multi-omic approach. This would need to include simultaneous ATACseq and ChIPseq to meet the challenges arising from the region-specificity of phenotypic rescue. This is further challenged by the limited available material for specific cranial neural tube regions, and the requirement for the inclusion of a tagged *Gli3* allele in the breeding scheme for ChIP (stemming from a lack of ChIP-specific anti-*GLI3* antibodies). Given the significant additional resource and time investment required, such studies lie beyond the scope of the current manuscript but we refer to these future directions and lay out a path forward in the limitations section of the Discussion.

-Clarification: The manuscript convincingly shows that *GLI3R* expression rescues apical constriction defects in the midbrain lateral cells. However, the overt exencephaly phenotype is described primarily in the forebrain (failure of anterior neuropore closure and absence of optic cups). As written, it was unclear to me how a midbrain-specific defect translates into a more anterior phenotype. Perhaps, the authors could explicitly articulate this connection (as depicted in Figure 9) more clearly, more specifically how midbrain apical constriction is the critical mechanical driver of cranial fold elevation and that its failure prevents anterior neuropore closure.

-Questions about midbrain specificity: Perhaps I misunderstood the paper, but it seems to me that both the presence of the apical constriction defects in the *Gpr161* mutants and the effects of the *GLI3R* expression rescue are specific to very limited regions of the developing midbrain. Can the authors speculate more as to why this might be?

Thank you for these comments, which in conjunction with comments from other reviewers points out an unintentional point of confusion in our original manuscript. As the reviewer rightly points out, the apical constriction defects in the *Gpr161* mutants and the effects of the *Gli3R* expression rescue are specific to the prospective developing midbrain. We have now expanded our discussion of our working models for how midbrain specific apical constriction defects could drive cranial-wide closure failure, suggesting that the steep curvature of the cephalic flexure between the forebrain and the midbrain is likely sensitive to a midbrain specific elevation defect, leading to a failure to either establish the forebrain/midbrain closure point located roughly above the flexure, or to stall fold apposition and fusion in response to a failure to gentle the sharpness of the flexure itself as elevation proceeds. To complement this discussion, we have provided an expanded summary figure (Fig 9B) showing the genesis and extent of closure defects in the various mutant lines we examine in the manuscript. Additionally, thanks to the reviewers' questions we have

reexamined the difference between fully and partially rescued exencephaly in *Gpr161*; *Gli3R* double mutants and suggest that this region-specificity could underlie the fact that partial rescues show enhanced FB but not MB closure.

-Minor issue: The gene name is inconsistently written as "Grp161" instead of "Gpr161" in several places and should be corrected.

We apologize for the oversight and have corrected this unintentional typo in 6 places.

Reviewer 2: This excellent new paper from extends the Brooks Lab's exciting recent work on the interplay of cilia-mediated HH signaling and the cell biology of neural tube closure in mammals, and in this new collaboration with the more signaling-focused Mukhopadhyay Lab, the paper represents what I hope will be the first of many more such studies. It's a perfect fit for Development, and I strongly recommend publication with only minor comments. The paper extends previous work exploring how dysregulation of cilia-mediated HH signaling evokes neural tube closure defects (NTDs) in the mouse. Here, they have performed a careful study of NTDs and signaling in mice with mutations in the HH transducer *Gpr161*, which notably is a known genetic risk factor for human NTDs. Using SEM and whole mount staining and sections, they explore both the knockout and also a knockin mutant that renders *Gpr161* unable to enter and cilia and signal via cAMP. They follow this with an equally careful study of the genetic interaction between these alleles and mutant in the *Gli2* activator and the *Gli3* repressor. They also explore the A/P and D/V patterning of the brains in all of these mice, finding quite specific patterns of disruption. Here the outcome is very clear: de-repression, not excessive activation, of HH targets cause the NTDs. This work alone, involving some onerous genetics and careful analysis of important proteins with clinical implications would make the paper worthy of Development. But the second half of the paper is even cooler. Shifting from genetic to cell biology, they use quantitative imaging to analyze the cell behaviors underlying these NTDs and find that these mice diverge in subtle but important ways from previously examined HH-related mutants. Notably, while the entire cranial neural tube fails to close, defects in apical constriction were observed only in the midbrain and not the forebrain. These findings are especially important because human NTDs are wildly heterogeneous disorders, and in these experiment the authors begin to model that complexity in mouse models. This represents a lot of work, and the data are to me very convincing. I urge the Editor to resist any calls by other reviewers asking for the intangible "more."

We thank the reviewer for sharing our enthusiasm for helping unravel the regional and mechanistic complexities of HH patterning and cranial closure.

Minor Comments:

1. It will be useful, I think, to be more explicit about the stats performed to compare histograms, in the text and the main figure. I was chiding the authors mentally until I finally noticed the little arrowheads in Fig 8 and Supp. Fig. 8. And was such an analysis done for Fig 7G, h, M? The differences seem wildly significant, but no arrowheads there...

Thank you for this important comment and inducement to improve our presentation. We have now added statistical testing to all distribution/histogram data. While the simplest test would be a Kolmogorov-Smirnov to compare the distributions of control and experimental conditions, this test cannot handle replicates (i.e. individual embryos) and given the high number of cell areas derived from even a single embryo, the test quickly becomes vastly overpowered and will determine even apparently random fluctuations between stage-matched control distributions as significant. We therefore instead used a two-way ANOVA with Sidak's correction for multiple comparisons to compare the mean values of each histogram bin between control and experimental conditions. In the revised figures we now highlighted all bins that are significant ($p < 0.05$) with arrowheads in all distributions diagrams or explicitly stated that no bins were significantly different. We have also enlarged the arrowheads to make them more obvious. Finally, to be as explicit about the statistical testing as possible, we now include details on all presented statistical tests (including the t-test and one-way ANOVAs we use in other analyses) in both figure legends and methods.

2. I think reviewers would benefit if the summary model in Fig. 9 included other HH pathway

mutants from previous papers (Gli or Ift mutants for example). This will help the uninitiated reader better grasp the complexity of genotype/phenotype in this interesting series of NTD mutants.

3. Along the same lines, for readers who may be expert in neural morphogenesis but less familiar with HH signaling and cilia, could a diagram be added that sketches the logic of HH/Gpr161/Gli signaling? This would especially be useful for distinguishing Gli2 and Gli3 effects.

We agree and thank the reviewer for these suggestions. We have addressed them in three ways. First, we have added a schematic of ciliary signaling and the role that GPR161, GLI2, and GLI3 play in HH transduction (Figure 2A) to motivate our analysis of the digenic mutants. Second, we have summarized the forebrain and hindbrain patterning data in *Gpr161* and *Gli* mutants in Figure 5D. Third, we have extensively revised Figure 9 to include both additional genotype/phenotype summaries (e.g. for monogenic *Gli2* mutants). In addition, we have provided a new set of schematics in Figure 9B that attempt to present the onset and progression of cranial closure phenotypes for these various genotypes.

Reviewer 3: In this substantial manuscript, Brooks et al use allelic series to disentangle the contributions of SHH signalling through activation versus repression of genes to regulation of neurulation and anterior neural patterning in mice. The paper is packed with fascinating findings, such as regional differences in apical constriction regulation between the forebrain and midbrain and inter-relation between neurulation and eye development - with anophthalmia and exencephaly being commonly concurrent in mouse models. The starting point for the work is previous studies by these and other authors showing that loss of *Gpr161* causes exencephaly, de-repression or over-activation of SHH signalling, and that anterior neuroepithelial cells with high SHH signalling do not apically constrict. The authors first compare the phenotype of *Gpr161* loss of function with a variant unable to localise to the cilia. They then combine *Gpr161* knockout alleles with loss of *Gli2* or of *Gli3R*. *Gli3R* deletion rescued the neurulation phenotype whereas loss of *Gli2* rescued neither neurulation nor over-expression of the SHH target *Ptch1* in exencephalic neural tissue.

We thank the reviewer for their enthusiasm for our work and agree that this paper presents many fascinating twists on our understanding of HH and cranial closure.

The authors are asked to consider the following points in decreasing order of consequence:

1. "However, *Gpr161mut1/mut1* embryos by E11 had externally closed cranial neural tube (Fig. 1C)... However, the *Gpr161mut1/mut1* embryos developed exencephaly at later" and "we note closure of the anterior neuropore despite an open midbrain-hindbrain neuropore at later embryonic stages." The description of the neural tube phenotypes in the cilia-excluded *Gpr161* model is not clear. Is this a model in which the anterior neural tube is fully closed (Closures 1-3 all form, HNP and ANP both close), or a model in which the HNP fails to close as suggested in the discussion? Failure of HNP closure is difficult to differentiate from other closure failures at late stages, when exencephaly is present. Do the authors have data on the proportion of embryos with open HNPs at different somite stages? A re-opening phenotype would be very exciting given the rarity of models. The authors may wish to discuss findings in McLaren et al Dec Cell 2025 which analyses post-closure roofplate mechanics.

We agree with the reviewer that "re-opening phenotype[s]" are a very interesting and relatively poorly understood aspect of neural tube closure, and we apologize for not being as clear on this important point as we hoped in the initial submission. Motivated by the reviewer's enthusiasm and our desire to be as explicit as possible, we have carefully reexamined our data on the cilia-excluded *Gpr161* mutant (*Gpr161^{mut1/mut1}*) in terms of phenotype by stage and present a summary of this data in a table in Figure 1. Overall, we find that open midbrain hindbrain neuropore is quite rare at E9.5, when initial closure completes, though we did observe it in 1 out of 11 embryos (~9%) (new Figure 1D and Figure S1B). We then see increasing incidences in successive stages of post-closure cranial neural tube growth at peaking at 64% of embryos at E12.5-13.5 (new Figure 1D). We also note that despite closure of the hindbrain neuropore at E9.5 stage, we do see structural abnormalities in the hindbrain roof plate (as shown in the images of horizontal cryosections of the cranial neural tube in Figure R1). We feel these data support a reopening

phenotype in at least some *Gpr161 mut1/mut1* embryos and note these possibilities in the discussion and Figure 9 (including comparison regarding tissue deformability (McLaren et al., 2025) and roof plate/zippering (Maniou et al., 2021)).

2. Do any of the models analysed have grossly observable spinal phenotypes? If not, that is worth stating to avoid others killing mice to test them in future (and as further evidence of regionalisation of mechanisms).

We agree with the reviewer that “regionalization of mechanisms” underlying neural tube phenotypes is an important theme in our studies, more so as we see clear differences in phenotypes in forebrain versus hindbrain in ventral neuroprogenitor patterning and midbrain versus forebrain in apical constriction of lateral cranial folds. As we highlighted in our original Discussion, the Finnell lab has previously demonstrated that spinal bifida phenotypes are present in *Gpr161* mutants and appear to be responsive to WNT signaling (Kim et al., 2023; Li et al., 2015) and the planar cell polarity effector FUZ (Kim et al., 2024). Looking superficially into the *Gli3R* expressing *Gpr161* ko embryos at E10.5-E11.0 by SEM, we mostly note superficial rescue in the closure defects of the caudal neural tube in *Gpr161* ko (Figure R2). However, more detailed characterizations in these lines are currently being carried out in an ongoing collaboration with Sung-Eun Kim and Rick Finnell. Such characterization will also require conditional approaches targeting caudal spinal tissues to avoid mid-gestational embryonic lethality of the *Gli3R* expressing *Gpr161* ko and *Gli3R* expressing *Gpr161*; *Gli2* double ko by E12 and E12.75, respectively. Therefore, we respectfully suggest that careful examination of the spinal phenotypes that encompass both externally visible and soft-tissue defects (Avagliano et al., 2019), are beyond the scope of the current manuscript.

3. Interpretation of the *Gli2* KO allelic series depends on evidence that it achieved reduction in SHH signalling in the *Gpr161*-null background. They state “unlike the spinal cord, lack of *Gli2* function was unable to restrict the excessive HH signaling promoted by *Gpr161* loss in the open cranial neural tube”. Spinal cord expression of *Ptch1* is not described in the preceding text, but Figure 2C does appear to show lower staining in the spinal neural tube of *Gpr161*-Ko/*Gli2*-Ko than in *Gpr161*-ko alone. Is this reproducible? Excenephalic tissue can very ‘sticky’ to in situ probes and is exposed to abnormal signalling from the amniotic fluid (also seen in figure 4B?): might the spinal expression pattern more meaningfully represent HH status?

As the reviewer suggests Figure 2C “does appear to show lower staining in the spinal neural tube of *Gpr161*-Ko/*Gli2*-Ko than in *Gpr161*-ko alone”. We have now included similar stage embryos from both these genotypes highlighting similar results with *Ptch1* in situ hybridization (Figure R3).

We also note that *Gli2* ko embryos showed reduced floorplate and ventral progenitor marker expression at this stage at all spinal cord levels in a previous study (Hwang et al., 2023). In the current manuscript, we show reduced floor plate and ventral progenitor expression in forebrain and hindbrain for *Gli2* ko and hindbrain of *Gpr161 Gli2* double ko embryos (Figure S2). However, we actually see ventral progenitor expansion in the forebrain of *Gpr161 Gli2* double ko embryos, consistent with the *Ptch1* overexpression data (Figure 5, Figure S2,3).

Thus, our experiments reproducibly show a reduction of HH signaling in caudal neural tube in *Gpr161 Gli2* double ko compared to *Gpr161* ko using various methods, including *Ptch1* in situ hybridization.

4. Rescue of the exencephaly phenotype by *Gli3R Δ 701/+* is striking. While not critical to the interpretation of the data - do the authors know whether there is a sex bias in the rescue (given the well-documented female excess of exencephaly)? The methods state that both males and females were analysis so if that data is available, it would be worth including, although it probably does not meet the ethical bar for killing more mice to obtain in isolation.

This is a very interesting point, as the reviewer rightly notes the sex bias in exencephaly. For this revision, we determined sex of the E10.5-E11.5 of embryo cohorts examined in Figure 3C, where we saw rescue of the exencephaly phenotype in *Gpr161 ko/ko*; *Gli3R^{D701/+}* embryos. Using this limited set, we were able to determine sexes of 16/18 embryos from the stored DNA but did not

notice overt sexual dimorphism in rescue of exencephaly, for e.g. among the E10.5-E11.5 cohorts, 5/11 rescued embryos were males, whereas 3/5 partially rescued were males. The data is shown in Figure R4 and added in legend of Figure 3. While the N-values here are too low to formally test mild sex bias in rescue, these data would argue that there is not an extreme bias.

5. What is more important for regulation of apical constriction needed for midbrain neural tube closure: the absolute level of HH signalling or its midline localisation? The E8.5 *in situ* shown suggest both change with the alleles used.

This is another very interesting point and intersects with a deeper question about whether the apical constriction phenotypes are downstream of fate transversion. From our data, we see that *Gpr161* ko starts to show high HH signaling in the *Ptch1* ISH assay only beginning at 7+ somites. As we see apical constriction starting to be affected at earlier stages (5-7 somite stage), we believe that derepression of HH targets intersecting with dorsolateral cell fate is likely not required *per se* for regulating apical constriction. Additionally in our previous work (Brooks et al., 2020) when we conditionally activated the HH response in a ligand independent way specifically in the midbrain using *Wnt1-Cre*, we noted that the impact on apical constriction appeared to be at least somewhat autonomous, as cells immediately adjacent to the expression domain apically constricted normally. Together, these observations would argue more strongly for a model of control by relative HH signaling levels than directly by spatial restriction of the signal. Designing experiments to disentangle these possibilities is difficult, even at a conceptual level, but are certainly worth pursuing in the future.

6. Figure 7H - the label above the horizontal line is mis-centred.

Apologies for the oversight, this has been corrected.

7. "Lack of *Gli2* and *Gli3R* expression..." sounds like a double knockout model.

Thanks for pointing this out, we have corrected the text to "Lack of *Gli2* or *Gli3R* expression"

8. Line 189: "First, we first"

Thank you, we have corrected this to "First, we tested".

References

- Avagliano, L., V. Massa, T.M. George, S. Qureshy, G.P. Bulfamante, and R.H. Finnell. 2019. Overview on neural tube defects: From development to physical characteristics. *Birth Defects Res.* 111:1455-1467.
- Brooks, E.R., A.R. Moorman, B. Bhattacharya, I.S. Prudhomme, M. Land, H.L. Alcorn, R. Sharma, D. Pe'er, and J.A. Zallen. 2025. A single-cell atlas of spatial and temporal gene expression in the mouse cranial neural plate. *Elife.* 13.
- Hwang, S.H., K.A. White, B.N. Somatilaka, B. Wang, and S. Mukhopadhyay. 2023. Context-dependent ciliary regulation of hedgehog pathway repression in tissue morphogenesis. *PLoS Genet.* 19:e1011028.
- Kim, S.E., P.J. Chothani, R. Shaik, W. Pollard, and R.H. Finnell. 2023. Pax3 lineage-specific deletion of *Gpr161* is associated with spinal neural tube and craniofacial malformations during embryonic development. *Dis Model Mech.* 16.
- Kim, S.E., H.Y. Kim, B.J. Wlodarczyk, and R.H. Finnell. 2024. Linkage between *Fuz* and *Gpr161* genes regulates sonic hedgehog signaling during mouse neural tube development. *Development.* 151.
- Li, B.I., P.G. Matteson, M.F. Ababon, A.Q. Nato, Jr., Y. Lin, V. Nanda, T.C. Matise, and J.H. Millonig. 2015. The orphan GPCR, *Gpr161*, regulates the retinoic acid and canonical Wnt pathways during neurulation. *Dev Biol.* 402:17-31.
- Maniou, E., M.F. Staddon, A.R. Marshall, N.D.E. Greene, A.J. Copp, S. Banerjee, and G.L. Galea. 2021. Hindbrain neuropore tissue geometry determines asymmetric cell-mediated closure dynamics in mouse embryos. *Proc Natl Acad Sci U S A.* 118.
- McLaren, S.B.P., S.L. Xue, S. Ding, A.K. Winkel, O. Baldwin, S. Dwarakacherla, K. Franze, E.

Hannezo, and F. Xiong. 2025. Differential tissue deformability underlies fluid pressure-driven shape divergence of the avian embryonic brain and spinal cord. *Dev Cell.* 60:2237-2247 e2234.

Figure R1

Figure R1. Hindbrain roof plate in *Gpr161* *mut1/mut1* embryos. Immunofluorescence images of horizontal cryosections of the cranial neural tube in wildtype and *Gpr161* *mut1/mut1* embryos immunostained for Pax6 and counterstained with DAPI are shown. Please note structural abnormalities (arrow) in the hindbrain roof plate in *Gpr161* *mut1/mut1* embryos. FB, forebrain; HB, hindbrain. Scale, 100 μ m.

Figure R2

Figure R2. Superficial rescue of spina bifida in *Gpr161* ko embryos by *Gli3R* expression. *En face* views of spinal regions of control, *Gpr161* ko and *Gpr161* ko; *Gli3R* embryos are shown. We mostly noted closed spinal neural tube irrespective of full or partial rescue of exencephaly.

Figure R3

Figure R3. *Ptch1* in situ hybridization controls. Whole mount *Ptch1* in situ hybridization in control, *Gpr161* ko/ko; *Gli2* ko/+ and *Gpr161* ko/ko; *Gli2* ko/ko at E9.5-E10. Please note lower *Ptch1* ISH staining in the spinal neural tube of *Gpr161* ko/ko; *Gli2* ko/ko than in *Gpr161* ko/ko; *Gli2*ko/+. Spina bifida is marked by arrow.

Figure R4

	Partially rescued		Rescued	
	Male	Female	Male	Female
E10.5-E11.0	2	2	2	3
E11.25-E11.5	1		3	3

Figure R4. Table showing sex of *Gpr161* ko embryos expressing *Gli3R* expression with respect to full or partial rescue of exencephaly phenotype.

Second decision letter

MS ID#: dev.205171R1

MS Title: GPR161-GLI3 repressor signaling at cilia directs apical constriction and cell fate during cranial neural tube closure

Authors: Eric R. Brooks; Sun-Hee Hwang; Kevin A. White; Saikat Mukhopadhyay

Article Type: Research Article

Dear Dr Mukhopadhyay,

I have now received all the referees reports on the above manuscript, and have reached a decision. The referees' comments are appended below.

The overall evaluation is positive and we would like to publish a revised manuscript in Development. Referee 3 has a few minor suggestions to improve clarity. Please adjust, as appropriate.

Please attend to all of the reviewers' comments in your revised manuscript and detail them in your point-by-point response. If you do not agree with any of their criticisms or suggestions explain clearly why this is so. If it would be helpful, you are welcome to contact us to discuss your revision in greater detail. Please send us a point-by-point response indicating your plans for addressing the referees' comments, and we will look over this and provide further guidance.

Reviewer 1

SUMMARY OF THE ADVANCE MADE IN THIS PAPER AND ITS POTENTIAL SIGNIFICANCE TO THE FIELD

Revisions look good. Thank you for making changes to the figures.

Reviewer 2

SUMMARY OF THE ADVANCE MADE IN THIS PAPER AND ITS POTENTIAL SIGNIFICANCE TO THE FIELD

The advances were outlined in my previous review.

SUGGESTIONS TO AUTHORS

My very minor concerns have all been addressed in the revision and strongly support publication.

Reviewer 3

The authors have made substantial improvements to the manuscript and provided compelling new data which fully address my previous comments. A few minor corrections are listed below:

- Line 164: "into the Gpr161 background" worth specifying this is the 'ko' background not the mut1 used above
- Line 266: "We next asked if Gpr161 mutants" - actually ko not the mutant 'mut1' used earlier in the paper
- Line 220: "in all genotypes Gpr161 ko and Gpr161; Gli2 double ko embryos" - remove 'genotypes'?
- Line 241: "in pathway logic between the forebrain that appears" - this sentence is confusing - a comparator is expected after forebrain (between the forebrain and hindbrain/spinal cord)
- New schematic in Figure 9: This is very useful. However, it does not show Closure 2 forming, which normally happens before the forebrain anterior neuropore closes. One assumes the authors are using a mouse strain which does make Closure 2? This distinction is likely also relevant to the rescue/lack of rescue. The Gpr161Ko/Ko show very flat neural folds with no evidence of elevation, consistent with an apical constriction failure. The 'partial rescue' embryos in Figure 3 show elevated neural folds which may have made Closure 2 - again consistent with a rescue of apical constriction. Elevation of the neural folds and approximation where Closure 2 should form is also suggested in some Gpr161Ko/Ko;Gli2Ko/Ko embryos shown e.g. Fig 2B, C - suggesting some rescue of constriction - although clearly insufficient to achieve closure. Amending the sequence to illustrate Closure 2 in the schematics will make them more interpretable.

Second revision

Author response to reviewers' comments

Thanks for the positive comments. We have now revised Figure 9 incorporating Reviewer 3's valuable comments. We thank all reviewers for improving the manuscript with their comments and suggestions! All changes are highlighted in text.

Comments from the Reviewers:

Reviewer 1: SUMMARY OF THE ADVANCE MADE IN THIS PAPER AND ITS POTENTIAL SIGNIFICANCE TO THE FIELD 4 of 5

Revisions look good. Thank you for making changes to the figures.

Thank you

Reviewer 2: SUMMARY OF THE ADVANCE MADE IN THIS PAPER AND ITS POTENTIAL SIGNIFICANCE TO THE FIELD

The advances were outlined in my previous review.

SUGGESTIONS TO AUTHORS My very minor concerns have all been addressed in the revision and strongly support publication.

Thank you

Reviewer 3: The authors have made substantial improvements to the manuscript and provided compelling new data which fully address my previous comments.

A few minor corrections are listed below:

- Line 164: "into the Gpr161 background" worth specifying this is the 'ko' background not the mut1 used above
- Line 266: "We next asked if Gpr161 mutants" - actually ko not the mutant 'mut1' used earlier in the paper
- Line 220: "in all genotypes Gpr161 ko and Gpr161; Gli2 double ko embryos" - remove 'genotypes'?
- Line 241: "in pathway logic between the forebrain that appears" - this sentence is confusing - a comparator is expected after forebrain (between the forebrain and hindbrain/spinal cord)

Thank you for your close reading. We have corrected or clarified the text at each position.

- New schematic in Figure 9: This is very useful. However, it does not show Closure 2 forming, which normally happens before the forebrain anterior neuropore closes. One assumes the authors are using a mouse strain which does make Closure 2? This distinction is likely also relevant to the rescue/lack of rescue. The Gpr161Ko/Ko show very flat neural folds with no evidence of elevation, consistent with an apical constriction failure. The 'partial rescue' embryos in Figure 3 show elevated neural folds which may have made Closure 2 - again consistent with a rescue of apical constriction. Elevation of the neural folds and approximation where Closure 2 should form is also suggested in some Gpr161Ko/Ko;Gli2Ko/Ko embryos shown e.g. Fig 2B, C - suggesting some rescue of constriction - although clearly insufficient to achieve closure. Amending the sequence to illustrate Closure 2 in the schematics will make them more interpretable.

Thank you for this suggestion to improve the summary closure schematics in Figure 9B. We originally had closure 2 only implicitly in the figure, and we agree this is an important point to make

explicitly as we do in the discussion text. We have refactored the figure to now show either correct formation or failure to form closure 2 in the relevant genotypes. We also amended the legend to reflect the nature of the arrows: black arrows mark elevation forces, whereas green and magenta arrows indicate functional or nonfunctional apposition/fusion activity, respectively.

Third decision letter

MS ID#: dev.205171R2

MS Title: GPR161-GLI3 repressor signaling at cilia directs apical constriction and cell fate during cranial neural tube closure

Authors: Eric R. Brooks; Sun-Hee Hwang; Kevin A. White; Saikat Mukhopadhyay
Article Type: Research Article

Dear Dr Mukhopadhyay,

I am happy to tell you that your manuscript has been accepted for publication in Development, pending our standard publication integrity checks.